# Condensin loop extrusion properties, roadblocks, and role in homology search during recombination in *S. cerevisiae*

Vinciane Piveteau [1,2], Chloé Dupont [1,2], Hossein Salari [1], Agnès Dumont[1], Jérôme Savocco [1], Daniel Jost [1] & Aurèle Piazza [1✉]

## Abstract

The in vivo mechanism, *cis*-acting roadblocks, and biological functions of DNA loop extrusion by eukaryotic SMC complexes remain incompletely defined. Here, we identify condensin-dependent Hi-C contact stripes at the recombination enhancer (*RE*) and at *rDNA* in *S. cerevisiae*. The *RE* is an autonomous condensin loading site only active in *MAT*a cells from which oriented, unidirectional loop extrusion proceeds with an estimated processivity ~150–250 kb and a density ~0.04–0.18 that varies across the cell cycle. Centromeres, replication forks, and highly transcribed RNA PolII-dependent genes represent roadblocks for condensin. Cohesin is not an obstacle for condensin, while Top2 promotes its loop extrusion activity. A DNA double-strand break (DSB) at *MAT* blocks loop extrusion, resulting in the establishment of a ~170 kb-long *RE-MAT* loop. The *RE* and the DSB are required and sufficient to form this site-specific loop, which promotes *RE*-proximal homology identification in the early stages of recombinational DNA break-repair. We propose that juxtaposition of the broken *MAT*a site and its target *HML*α donor is the relevant structure by which condensin promotes a-to-α mating-type switching.

**Keywords** Condensin; Loop extrusion; SMC; Homology search; Mating-type switching
**Subject Category** Chromatin, Transcription & Genomics

## Introduction

Most Structural Maintenance of Chromosome (SMC) family of ATPase complexes, including cohesin, condensin, and Smc5/6, share the ability to extrude DNA loops in vitro presumably via a conserved mechanism (Ganji et al, 2018; Davidson et al, 2019; Kim et al, 2019; Kong et al, 2020; Kim et al, 2020; Golfier et al, 2020; Pradhan et al, 2022; Shaltiel et al, 2022; Ryu et al, 2022; Pradhan et al, 2023; Janissen et al, 2024). Studying the function and activities of these complexes in cells

has proven more challenging, as they generally do not have well-defined loading sites from which extrusion can be experimentally tracked. This is at the notable exception of the condensin-like bacterial SMC loading at *parS* sites (Gruber and Errington, 2009; Sullivan et al, 2009), and of a specialized condensin complex (condensin DC) loaded at *rex* sites on the X chromosome in XX hermaphrodites *C. elegans* animals (Morao et al, 2022), which enabled studying the loop extrusion activities of these SMCs and identify roadblocks in vivo with Hi-C (Morao et al, 2022; Brandão et al, 2019, 2021; Wang et al, 2017; Tran et al, 2017).

Condensin I and II are conserved SMC complexes specialized in chromosome assembly and segregation (Hirano, 2016). In *S. cerevisiae*, a single condensin complex individualizes mitotic chromosomes and promotes rDNA segregation at anaphase (Lazar-Stefanita et al, 2017; Renshaw et al, 2010; Guérin et al, 2019; Freeman et al, 2000). It is also associated with chromatin at all cell cycle phases, suggesting roles outside of mitosis (Freeman et al, 2000; D'Ambrosio et al, 2008; Leonard et al, 2015). Recently, it has been involved in mating-type (*MAT*) switching (Li et al, 2019; Dinda et al, 2023), a genetic event induced right prior to or upon S-phase entry and that takes ~1 h to complete (Hicks et al, 1977; Nasmyth, 1983; Connolly et al, 1988). Mating-type (*MAT*) switching is a specialized recombination-dependent process that must select between two competing silent donor loci. As such, it represents a classic model for studying the mechanisms underlying the establishment of specific interactions along chromosomes. *MAT* switching is initiated by a site-specific DNA double-strand break (DSB) inflicted by the homothallic (HO) endonuclease at the *MAT* locus on chr. III (Haber, 2012). Gene conversion occurs upon repair by homologous recombination using one of two silent donors carrying the opposite mating-type information as a template: *HML*α and *HMR*a at the left and right extremity of chr. III, respectively. Of note, *MAT* is located ~95 kb away from *HMR*a on the same chromosome arm, yet *MAT*a cells efficiently use the more distant *HML*α donor present ~186 kb away across the centromere. Donor preference is regulated by a bipartite "recombination enhancer" (*RE*) element present ~14 kb away from *HML*α, only active in *MAT*a cells (Wu and Haber, 1996). The first part binds the Rad51-ssDNA filament in *trans* and promotes homology search in its vicinity (Renkawitz et al, 2013; Dumont et al, 2024), and the second part is required for the horseshoe folding of chr. III,

[1]Université de Lyon, ENS de Lyon, Université Claude Bernard, CNRS UMR5239, Laboratoire de Biologie et Modélisation de la Cellule, Lyon, France. [2]These authors contributed equally: Vinciane Piveteau, Chloé Dupont. ✉E-mail: aurele.piazza@ens-lyon.fr

that brings the left and right chromosomal arms in close proximity (Belton et al, 2015; Li et al, 2019). This *MAT*a-specific horseshoe folding depends on condensin, which is enriched at the *RE* in a Sir2-, Tof2-, Fob1-, cohibin-, and Mcm1-dependent manner, suggesting that it is loaded there (Li et al, 2019; Dinda et al, 2023). Accordingly, condensin and the proteins required for its enrichment at the *RE* promote usage of *HML*α upon DSB formation at *MAT*a (Li et al, 2019; Dinda et al, 2023). Such a non-mitotic role of condensin has been ascribed to its chr. III folding activity. A second site enriched for condensin is the replication fork barrier (*RFB*) site within *rDNA* repeats, with largely overlapping requirements with the *RE* at the notable exception of the Matα2-repressible Mcm1 factor (Johzuka and Horiuchi, 2009). These putative loading sites represent an opportunity to study the loop-forming activity of condensin in eukaryotic cells and its function in establishing specific chromosomal interactions.

Here, using Hi-C, we show the existence of discrete condensin-dependent contact stripes emanating from the *RE* and the *rDNA* locus. By analyzing these stripes, we provide evidence that condensin unidirectionally extrudes chromatin loops from these sites with a defined orientation in cells. In conjunction with a polymer model for loop extrusion, we exploit this system to infer the basic loop extrusion properties of condensin in vivo and delineate its regulation in *cis* by various types of roadblocks. Functionally, condensin loop extrusion properties are uniquely exploited during *MAT*a-to-α switching, upon DSB-dependent reconfiguration of chr. III structure.

## Results

### Two condensin-dependent contact stripes in the budding yeast genome

Chromatin immunoprecipitation (ChIP)-based assays revealed that the *RE* is the unique genomic region most enriched for condensin in *MAT*a cells (Fig. 1A) (Li et al, 2019; Costantino et al, 2020; Rossi et al, 2021). Such approaches cannot straightforwardly disambiguate loading from accumulation sites of long-range translocases such as SMCs, nor do they give information about their activity, unlike Hi-C (Wang et al, 2017; Brandão et al, 2021; Wike et al, 2021; Guo et al, 2022; Morao et al, 2022; Kim et al, 2025). Previous Hi-C experiments failed to establish a clear contact signature at the *RE* (Duan et al, 2010; Belton et al, 2015; Li et al, 2019) while recent micro-C data indicated the presence of a contact stripe (Dinda et al, 2023). Our Hi-C protocol that yields contact maps at sub-kilobase resolution was applied to an asynchronous population of haploid *MAT*a and *MAT*α cells. It confirmed the presence of a discrete, *MAT*a-specific contact stripe emanating from the *RE* and stretching across chr. III (Fig. 1A). It depended on the presence of the *RE*, as did the overall chr. III folding (Fig. EV1A) (Belton et al, 2015). This stripe could be visualized as a 4C-like profile and compared to control sites in the yeast genome (Fig. 1B and see "Methods" for the selection of control sites), which revealed a specific enrichment of contacts from 40 kb to the end of chr. III. These observations indicated that a heterogeneous population of loops links the *RE* and the rest of chr. III specifically in *MAT*a cells (Belton et al, 2015; Li et al, 2019; Dinda et al, 2023; Li et al, 2024). Similar to the *RE*, a contact stripe could be observed emanating from

the left rDNA-flanking region and extending up to the centromere of chr. XII in both mating types (Fig. EV1B).

We addressed the dependency of these contact stripes on condensin by depleting its Smc2 subunit using the auxin-inducible degron (AID) system (Morawska and Ulrich, 2013) in an asynchronous culture of haploid *MAT*a cells (Figs. 1C and EV1C). Condensin depletion did not affect the cell cycle distribution, the genome-wide probability of contact $P_c$ as a function of genomic distance $s$, and only exhibited substantial changes in contacts on chr. III and XII (Fig. EV1C–E) (Lazar-Stefanita et al, 2017; Costantino et al, 2020). The associated stripe on chr. III was lost upon condensin depletion (Figs. 1C and EV1F). The contact stripe emanating from the rDNA-flanking region and extending up to the centromere of chr. XII also depended on condensin (Figs. 1D and EV1E). No corresponding stripe was found on the telomere side of the rDNA (Fig. EV1E), suggesting that loop formation by condensin is oriented. No other condensin-dependent stripe was detected genome-wide.

These results show that condensin forms two heterogeneous population of loops with discrete anchors on chr. III and XII. In particular, condensin folds chr. III has a heterogenous population of loops from the *RE* specifically in *MAT*a cells (Fig. 1E), corroborating prior suggestions (Li et al, 2019; Dinda et al, 2023). *MAT*α cells thus provide a physiological "off" state for condensin specifically at the *RE*, which we exploited in the following sections.

### The *RE* is a directional and oriented condensin loop anchor

The *RE* contains two main modules: a left Fkh1-binding region and a right condensin-binding region (Fig. 2A) (Li et al, 2019; Haber, 2012). In order (i) to address whether the *RE* was sufficient to form a contact stripe outside of the context of chr. III, (ii) to determine its reliance on the Fkh1 module, and (iii) to disambiguate the stripe signal from that of the brush configuration conferred by the Rabl chromosome organization (Duan et al, 2010), we introduced the full *RE* or a truncated version lacking the Fkh1 module (*RE-right*) at an interstitial location in the longest, rDNA-devoid, chromosome arm of the yeast genome (right arm of chr. IV, Figs. 2A,B and EV2A). It resulted in the formation of a single, *MAT*a-specific stripe emanating from the *RE* or *RE-right* constructs that extended up to the right end of chr. IV, several hundreds of kilobases away (Figs. 2B,C and EV2A). The direction of the stripe could be reversed upon inversion of the *RE* constructs (Figs. 2B,D and EV2A). Hence, the *RE* is required and sufficient for forming a discrete contact stripe exhibiting a specific orientation in *MAT*a cells, independently of its Fkh1-binding module.

These observations provide strong evidence that condensin can extrude chromatin loops in *S. cerevisiae* cells. These loops have two well-defined anchors that correspond to highly enriched condensin binding sites: the *RE* and at the *rDNA*. They may originate from an unidirectional loop extrusion process initiated at the site present at the base of the stripe, or from a site-specific block of loop extrusion events initiated at multiple dispersed sites (Fudenberg et al, 2016, 2017; Vian et al, 2018). In the following result sections, we assumed the first scenario to be correct. This assumption is supported by additional evidence presented below and summarized in "Discussion".

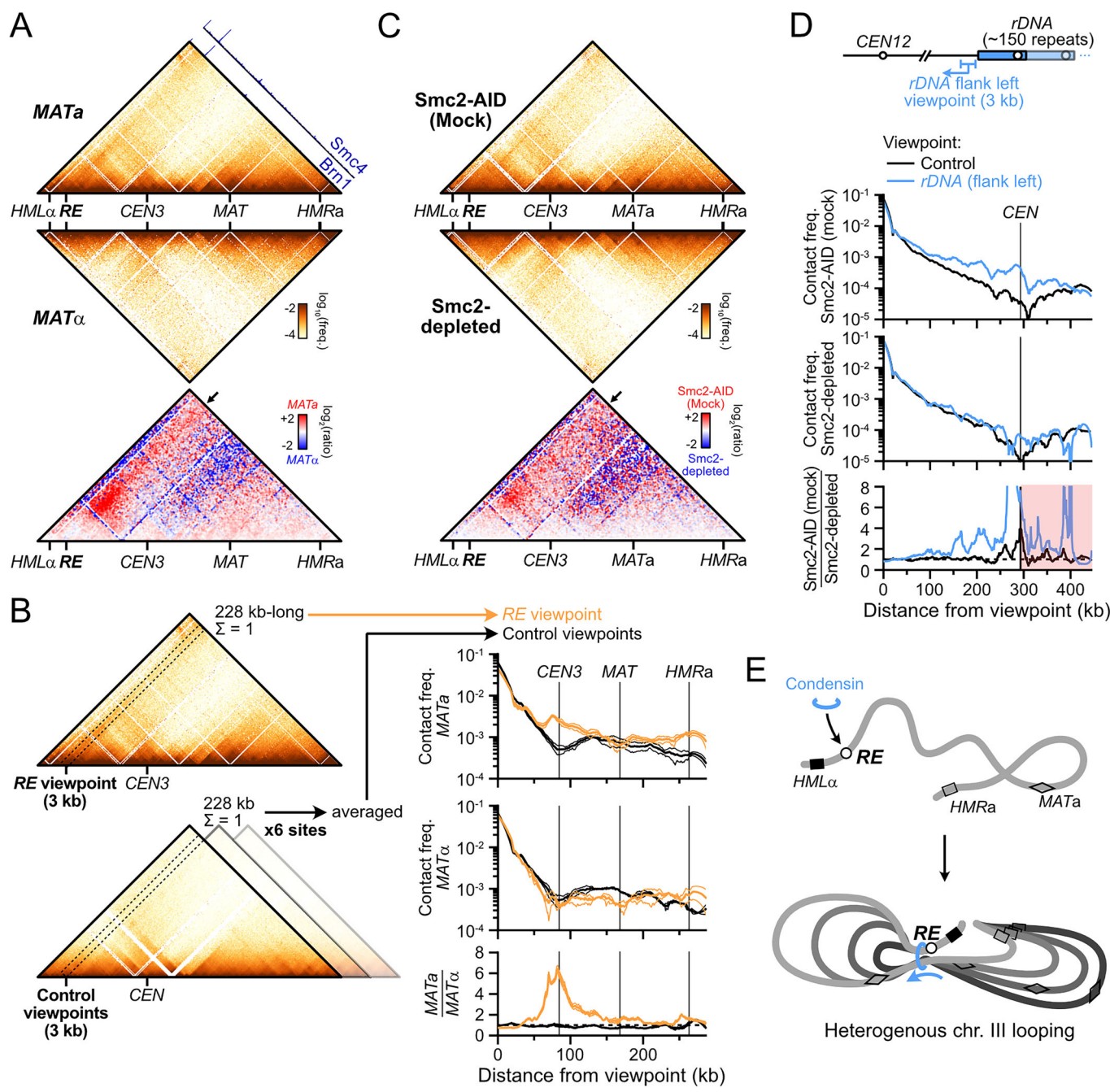

**Figure 1. Two condensin-dependent Hi-C contact stripes in the budding yeast genome.**

(**A**) Hi-C contact maps (top) and ratio map (bottom) of chr. III in exponential cultures of *MATa* (APY142) and *MATα* (APY295) cells. Data show the merge of n = 4 and n = 2 biological replicates, respectively. Calibrated ChIP-Exo profiles are from ref. (Rossi et al, 2021). Bin: 1 kb. (**B**) Left: Rationale for the computation of *RE* 4C-like profiles and comparison to equivalent control sites (for precise genomic coordinates, see "Methods"). Right: 4C-like contact profiles of the *RE* and of the average of 6 control sites in *MATa* and *MATα* cells, from Hi-C data in (**A**). Bottom: Ratio of *MATa* and *MATα* profiles. Data show mean ± SEM of n = 4 and 2 biological replicates, respectively. (**C**) Hi-C contact maps and ratio maps of Smc2-AID and Smc2-depleted *MATa* cells (YTG155). Bin: 1 kb. (**D**) 4C-like profiles with the left rDNA-flanking region as a viewpoint, and the average of 6 corresponding control sites. Data show n = 1 biological replicate. (**E**) Model for *RE*-dependent chr. III loop folding by condensin in *MATa* cells.

# Density and processivity of loop extrusion by condensin inferred from biophysical modeling

We sought to infer properties of loop extrusion by condensin in cells by comparing experimental contact data with the output of a quantitative "scrunching" framework (Fig. 2E). It integrates the

unidimensional stochastic motion of loop extruders along the chromosome into a simple polymer model to predict the impact of loop extrusion on contact frequencies, which can be compared to Hi-C contact data (Alipour and Marko, 2012; Sanborn et al, 2015; Fudenberg et al, 2016; Chan and Rubinstein, 2023). In this framework, condensin can bind and unbind from its loading site at

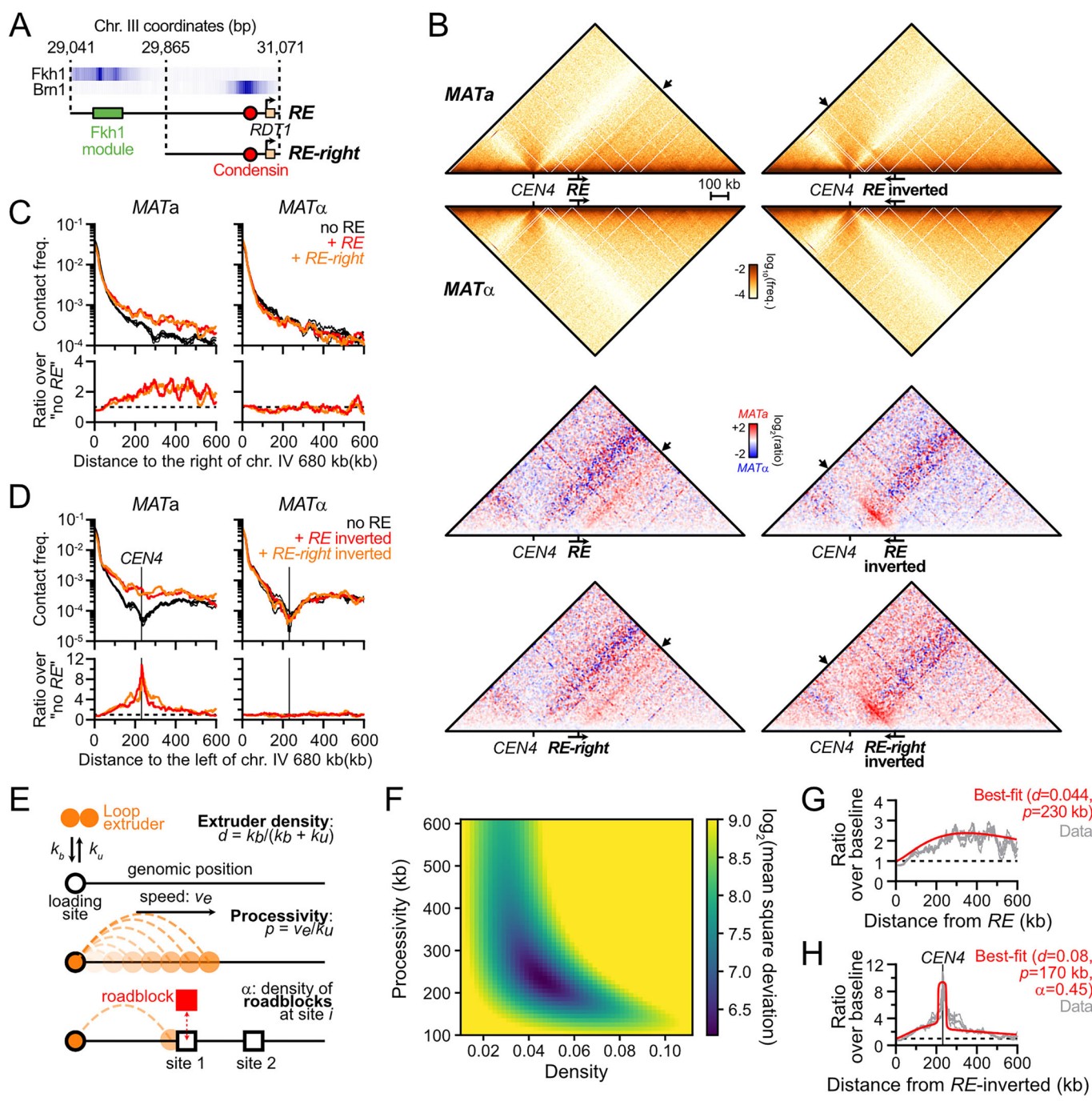

**Figure 2. Condensin loop extrusion properties.**

(A) Description of the *RE* and *RE-right* constructs introduced at position 680,258 on chr. IV. ChIP profiles are from (Rossi et al, 2021). (B) The *RE* is autonomous in forming a single oriented contact stripe. Top: Hi-C contact maps of chr. IV in *MAT*a and *MAT*α cells bearing the *RE* at position 680,258 either in the forward (APY1846 and APY1848) or inverted (APY1911 and APY1913) orientation. Bottom: Ratio maps of *MAT*a over *MAT*α cells in cells bearing either the *RE* or *RE-right* in the forward or the inverted orientation. The *RE-right* data are from Fig. EV2A. Hi-C maps are binned at 5 kb. Ratio maps are binned at 10 kb. (C) Top: 4C-like contact profiles of the chr. IV 680 kb coordinate either unmodified, or upon insertion of the *RE* or the *RE-right* constructs in *MAT*a (APY266, APY1846 and APY1850) and *MAT*α cells (APY295, APY1848 and APY1852). Bottom: Ratio of the 4C-like profiles with a *RE* construct over the "no *RE*" baseline. Data show the mean ± SEM of n = 4 and 2 biological replicates for the "no *RE*" *MAT*a and *MAT*α cells, respectively. n = 1 biological replicate for each +*RE* constructs. (D) Same as (C) with *RE* constructs in the inverted orientation (APY1848, APY1852, APY2058, and APY2060), and 4C profiles oriented towards the left end-side of chr. IV. (E) Rationale of the unidirectional loop extrusion crunching model, parameters, and output. (F) Heatmap of the mean squared deviation (MSD) between theoretical predictions and experimental data, expressed as the ratio of Hi-C interactions of the *RE* and *RE-right* constructs inserted on chr. IV in the forward direction versus no *RE* in *MAT*a cells (see Fig. EV2B). The region with the lowest MSD corresponds to the best-fit of processivity and loop density. (G) Observed data and best-fit simulated contact frequencies. Data are the average of the *RE* and *RE-right* constructs in the forward orientation in *MAT*a cells from (C). (H) Same as (C) with *RE*-inverted and *RE-right*-inverted construct in *MAT*a cells (data from (D)).

rates $k_b$ and $k_u$, respectively. Upon binding, one leg of the condensin remains anchored at the binding site while the other translocates along the chromosome at a speed $v_e$. These parameters define two key observables: the processivity $p = v_e/k_u$, which describes the average loop size extruded by condensin in the absence of a roadblock; and the density of loop extruders bound to DNA at any single time $d = k_b/(k_b + k_u)$. Pausing at site-specific roadblocks $i$ can also be modeled. This allows one to estimate the probability of observing an extruding loop between the condensin loading site and any other site along the chromosome and thus to predict their contact frequency. We first applied our framework to infer model parameters consistent with the stripe emanating from the *RE* and *RE-right* constructs inserted on chr. IV that lacks discernible loop extrusion roadblocks (see below). This led to best-fitting parameters for condensin processivity $P$ in the ~170–250 kb range, with a density of condensin on chromatin $d$ in the ~0.04–0.06 range (Figs. 2F,G and EV2B). Consistently, loop extrusion proceeded with a processivity $P = 170$ kb and a density $d = 0.08$ from the inverted *RE* constructs, and revealed a pause frequency ~0.45 at the centromere (Fig. 2H). Condensin-mediated loop extrusion proceeded similarly from its endogenous loading sites, with a processivity $P = 120$ kb and a density $d = 0.11$ from the *rDNA* and $P = 150$ kb and a density $d = 0.06$ from the *RE* in asynchronous cells. Consistently, the *SMC2-AID* control strain yielded similar $P = 150–170$ kb and $d = 0.05–0.07$, while Smc2 depletion caused a reduction of $d = 0.02$ at both sites without affecting the processivity. These results indicate that condensin is a chromatin loop extruder with a processivity ~three- to ninefold greater than that estimated for cohesin in mitotic and meiotic *S. cerevisiae* cells (Schalbetter et al, 2017, 2019).

## The density and processivity of condensin-dependent loops are cell cycle-regulated

Although budding yeast condensin is detected on chromatin at all cell-cycle stages, its enrichment at specific genomic sites and its role in chromatin compaction and *rDNA-CEN12* tethering vary along the cell cycle (Lazar-Stefanita et al, 2017; Freeman et al, 2000; D'Ambrosio et al, 2008; Leonard et al, 2015; Bhalla et al, 2002). Consequently, we analyzed the loop extrusion by condensin on chr. III and XII in G1, S-phase, and in cells arrested in G2/M either artificially (i.e., repression of *CDC20* or addition of nocodazole) or upon activation of the DNA damage checkpoint (i.e., formation of an unrepairable site-specific DNA double-strand break on chr. V), and determined best-fitting loop extrusion parameters for each condition (Figs. 3A–E and EV3A–F). The contact stripes emanating from the *RE* on chr. III and the rDNA on chr. XII were detectable at all cell cycle stages, with phase-specific variations in intensity and pausing at centromeres (Figs. 3A–E and EV3B–F, and see below).

Cells arrested in G1 with alpha-factor exhibited a stripe signal similar to that of asynchronous cells on both chr. III and chr. XII (Figs. 3A and EV3B). Consistently, best-fit parameters yielded $P = 170$ kb and $d = 0.11$ for the *rDNA* and $P = 120$ kb and $d = 0.05$ for the *RE*. Differently, the contact stripe on chr. III during S-phase failed to progress past ~120 kb (Figs. 3B and EV3C), resulting in an altered chr. III conformation in which contacts between the left arm and the right arm distal to *MAT* strongly decreased (Fig. EV3G). Likewise, the condensin extrusion span on chr. XII did not exceed ~80 kb. Simulations indicated a reduction of processivity ($P = 80$ kb

and 30 kb at the *RE* and the *rDNA*, respectively) while loop density remained unchanged ($d = 0.09–0.1$). Hence, condensin translocation, but not loading, is inhibited during S-phase. Finally, the contact stripes were overall similar in G2/M and asynchronous cells on both chr. III and XII (Figs. 3C–E and EV3D–F): the estimated processivity remained constant ($P = 170$ kb) while loop density $d$ increased modestly at the *rDNA*, between 0.12 and 0.18 from 0.11 in G1. Hence, loop extrusion by condensin at these specific sites is active in mitosis prior to anaphase and presumably regulated at the loading stage, while its overall loop extrusion activity remains unaltered (Lazar-Stefanita et al, 2017; Guérin et al, 2019; Freeman et al, 2000; Leonard et al, 2015; Strunnikov et al, 1995).

## Replication forks stall loop extrusion by condensin

The condensin loop extrusion span was strongly reduced on both chr. III and XII during S-phase (Fig. 3B). *RE*-bound loop extrusion initiates in an early-replicated region and progressed on chr. III up to the broad vicinity of *ARS310*, which corresponds to the first late-replicating region encountered (Fig. 3F). Differently, the condensin loop extrusion collapsed before encountering the first early-replicating *ARS1215* region immediately on chr. XII. In order to address whether replication forks are roadblocks for condensin, we depleted the DNA replication initiation factor Cdc45 in G1-arrested cells and released them in S-phase (Fig. 3G). Preventing origins from firing partially restored condensin progression up to the end of chr. III and over ~200 kb on chr. XII, although not to the levels observed in G1 and mitosis (Fig. 3G). In contrast, authorizing replication firing but immediately blocking replication forks' progression at high hydroxyurea (HU) concentration further reduced the condensin loop extrusion span (Figs. 3H and EV3H,I). The immediate blocking of replication forks upon HU treatment further allowed visualizing discrete loops between the *RE* and replication origins (Fig. 3I). These observations indicate that replication forks are strong condensin translocation roadblocks, independently of replisome progression. The incomplete restoration of condensin loop extrusion upon Cdc45 depletion further suggests that DDK-primed replisome components can also act as a roadblock for condensin, independently of the establishment of replication forks.

## Condensin-dependent loops are shortened in the absence of Top2

Top2 promotes loop extrusion by Condensin DC in *C. elegans* (Morao et al, 2022), and condensin function in chromosome compaction is tightly coupled to Top2 activity in bacteria and eukaryotes (Hirano, 2016). We thus addressed whether condensin translocation requires Top2 activity by re-analyzing published Hi-C datasets of cells arrested in mitosis upon treatment with benomyl in which Top2 was either conditionally inactivated at restrictive temperature (*top2-4* (Holm et al, 1985)) or depleted (Top2-AID) upon auxin addition prior to S-phase entry and up to mitotic arrest (Jeppsson et al, 2024) (Fig. EV4A–D). The span of loop extrusion by condensin was reduced to ~80 kb on chr. III and seemingly abolished on chr. XII in the absence of Top2 activity (Fig. EV4A–D). The phenotype was slightly more penetrant upon Top2 heat-inactivation than Top2 depletion. These results suggest that Top2 promotes loop extrusion by condensin, and that the extent of this effect is locus-specific.

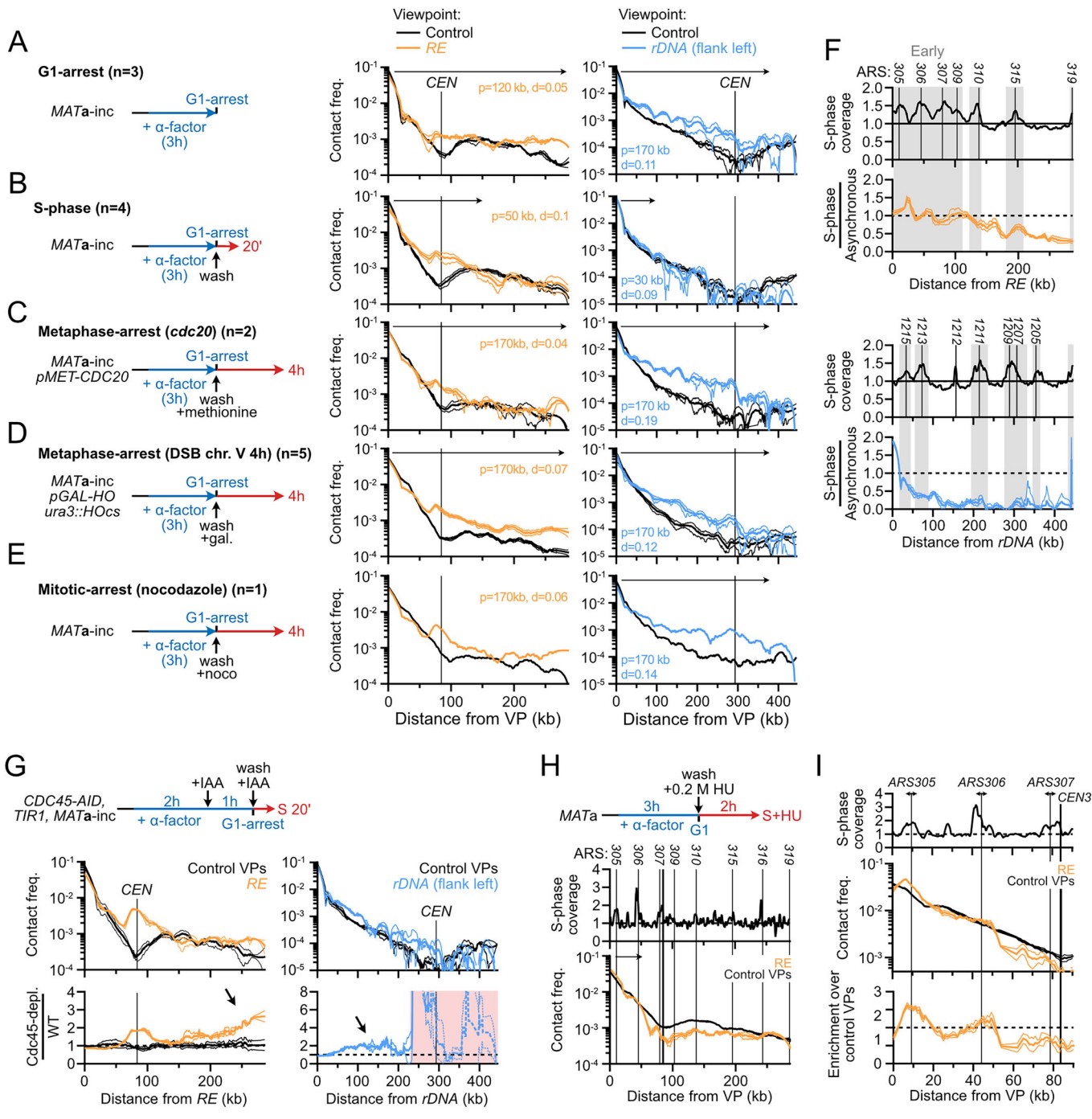

**Figure 3. Loop extrusion by condensin is inhibited by replication forks.**

(A–E) *RE* and *rDNA*-flanking 4C-like contact profiles (A) upon G1-arrest (APY266), (B) during S-phase (APY539 and APY607, merged), (C) upon metaphase-arrest due to *CDC20* repression (APY537), (D) upon DDC-induced metaphase-arrest due to formation of a single unrepairable HO-induced DSB at *ura3* (APY266), and (E) upon mitotic-arrest in the presence of nocodazole (APY266). All cells are *MAT*a. Best-fitting simulation processivity (*p*) and density (*d*) values are indicated. Data show mean ± SEM. The number of biological replicates n is indicated on each panel. (F) Top: Coverage from Hi-C reads in S-phase. Bottom: Ratio of the contact stripes in S-phase over the average of asynchronous cultures at the *RE* (top) and left rDNA-flanking (bottom) viewpoints. From data in (B). (G) Entry in S-phase in the absence of replication origin firing (Cdc45-AID + IAA) partially restores loop extrusion by condensin on chr. III and XII (APY539). Data show mean ± SEM of n = 2 biological replicates from (D'Asaro et al, 2025). (H) Stopping replisome progression immediately after S-phase entry prevents loop extrusion by condensin. Data show mean ± SEM of n = 3 biological replicates from (Jeppsson et al, 2022). (I) Zoom on data in (H).

## Determinants of the condensin roadblock at centromeres

To gain insights into the nature and determinants of the condensin loop extrusion pausing at centromeres (e.g., Figs. 1A and 2B,H), we related the roadblock at *CEN3* to the folding of the peri-centromeric regions across the cell cycle (Fig. 4A). No clear roadblock was detected in G1, consistent with the lack of condensin enrichment at peri-centromeres at that cell cycle stage (Leonard et al, 2015). A narrow roadblock appeared in S-phase, and became highly prominent in metaphase-arrested cells. This roadblock at the centromere was still observed upon treatment of cells with the microtubule-depolymerizing drug nocodazole, indicating it is independent of the mitotic spindle and of centromere clustering (Fig. 4B). Furthermore, Cdc45-depleted cells that reach metaphase in the absence of replication, and thus lack the sister chromatid required for the spindle to exert tension, also retained a localized roadblock (Fig. 4C). Roadblock appearance coincided with S-phase, when loop-extruding cohesins are loaded (Costantino et al, 2020; Dauban et al, 2020) and restructure the peri-centromere (Yeh et al, 2008; Paldi et al, 2020). The role of cohesin in blocking condensin was addressed by re-analyzing S-phase and metaphase Hi-C data obtained after Scc1 depletion in G1-arrested *MAT*a cells (Fig. EV5A,B) (Dumont et al, 2024; D'Asaro et al, 2025). The condensin roadblock persisted at the centromere (Figs. 4D and EV5C). More generally, condensin-dependent stripes appeared unaffected upon Scc1 depletion, indicating that condensin is oblivious to the presence of cohesin on chromatin (Fig. EV5A,B). Consequently, the centromeric roadblock for condensin is independent of spindle tension, centromere clustering, the centromeric chromatin hairpin, cohesin, and centromere replication.

We then addressed whether the centromeric function was required to block condensin in *cis*. To this end, we specifically inactivated the centromere of chr. III upon inducible transcription across *CEN3* using the *pGAL-CEN* system (Fig. 4E) (Hill and Bloom, 1987). Such a system causes the loss of the centromere-specific Cse4-containing nucleosome, and thus of the kinetochore, from *CEN3* (Nakabayashi and Seki, 2022). *CEN2* inactivation was used as a control (Fig. EV5D,E). Strong transcriptional activation for 2 h upon galactose addition caused a specific ~fourfold loss of contacts between the transcribed centromere and the 15 other centromeres, which was not restored after a 10 min transcriptional shutoff with glucose (Fig. EV5D). The appearance and disappearance of the border pattern typical of highly transcribed genes (Banigan et al, 2023; Salari et al, 2024; Chapard et al, 2025) confirmed the efficient transcriptional activation and shutoff (Fig. EV5E). The centromeric roadblock was abolished upon *CEN3* transcription and rapidly recovered following transcriptional shutoff (Fig. 4E). Transcription across *CEN2* did not affect the roadblock at *CEN3* (Fig. EV5F), indicating that the effect observed upon *CEN3* inactivation occurs in *cis*. Furthermore, the roadblock was lost in a strain defective for the dispensable outer kinetochore component Chl4, both in the presence and absence of spindle tension (Fig. 4F). These results indicate that the condensin roadblock at the centromere depends on the presence of the kinetochore in *cis*, irrespective of its role in mediating spindle tension and centromere clustering.

## Highly transcribed RNA PolII genes transiently stall condensin translocation

Besides centromeres, the contact stripes on chr. III and XII exhibited local contact variations suggestive of the presence of additional roadblocks along chromosome arms, which could be highlighted upon detrending of the contact stripes (Fig. EV6A). These positions were compared to various genomic and chromatin features detected by ChIP-Exo conducted in rich, glucose-containing media (Rossi et al, 2021) (Fig. 5A). Clearly, loop extrusion paused immediately ahead of sites enriched for RNA PolII, which exhibit the conserved "boundary" Hi-C pattern typical of highly transcribed genes (Fig. 5A) (Banigan et al, 2023; Salari et al, 2024; Chapard et al, 2025). These sites were also enriched for condensin, consistent with it pausing at highly transcribed genes (Fig. 5A). These pause profiles were also observed in G1- and metaphase-arrested cells (Fig. EV6B). The pause site ~160 kb away from the rDNA and not associated with an RNA PolII-enriched site (Fig. 5A) corresponded to the region immediately upstream of the *GAL2* gene, specifically activated in our galactose-containing culture conditions (Fig. 5A,B; galactose did not change the expression of the other genes in the *CEN12-rDNA* interval (see (Pelechano et al, 2013) and Fig. EV6D below). Accordingly, the boundary Hi-C pattern is readily detected at *GAL2* and other *GAL* genes (Figs. 5A,B and EV6C). Cells grown in the absence of galactose did not exhibit the boundary nor the increased *rDNA-GAL2* looping (Figs. 5B,C and EV6D). Transcriptional induction by the addition of galactose for only 10 min was sufficient to induce transcription of the *GAL* genes (Brouwer et al, 2023) and enrich for the *rDNA-GAL2* loop (Figs. 5C and EV6D,E). Conversely, repression of *GAL* genes upon glucose addition to galactose-containing media for 10 min (Johnston et al, 1994) caused a partial loss of the *rDNA-GAL2* loop (Figs. 5D and EV6F). Hence, condensin pausing ahead of a highly transcribed PolII gene can be rapidly established and reversed, suggesting that it is a primary consequence of its transcriptional activity. Pausing occurred irrespective of the gene orientation relative to the directionality of loop extrusion (e.g., *PDC1* and *PGK1* vs. *AHP1* and *GAL2* in Fig. 5A).

Inactivation of RNA PolII, either upon heat-inactivation of its Rpb1 subunit or upon thiolutin treatment, exerted profound effects on chromosome organization (Jeppsson et al, 2022), which prevented us from determining the role of transcription in regulating the processivity of loop extrusion by condensin over large chromosomal segments.

We further noted the presence of a pause site ~120 kb from the rDNA locus (position ~330 kb) not associated with a high occupancy of RNA PolII or PolIII (Figs. 5A and EV6B). The nature of this roadblock remains to be determined.

## Rad51-independent reconfiguration of chr. III structure upon DSB at *MAT*

What is the functional significance of the condensin-mediated loop extrusion spanning across chr. III? Previous reports revealed that condensin promoted usage of the *RE*-proximal *HMLα* donor during *MAT* switching in *MAT*a cells, which was presumed to originate from its role in chr. III folding (Li et al, 2019; Dinda et al, 2023). Differently, we sought to address whether a DSB induced at *MAT* may act as a condensin roadblock, with the resulting *RE-MAT* loop directly juxtaposing the DSB to the *HMLα* donor. To this end, we induced *HO* overexpression from a galactose-inducible promoter in asynchronous *MAT*a and *MAT*α cells. Repair-deficient *rad51Δ* mutants were used to (i) ensure the maintenance of a homogeneous population of *MAT*a or *MAT*α cells (i.e., no switching), and (ii) prevent the Rad51-dependent preferential recruitment of the DSB to the *RE* in *trans* (Renkawitz et al, 2013;

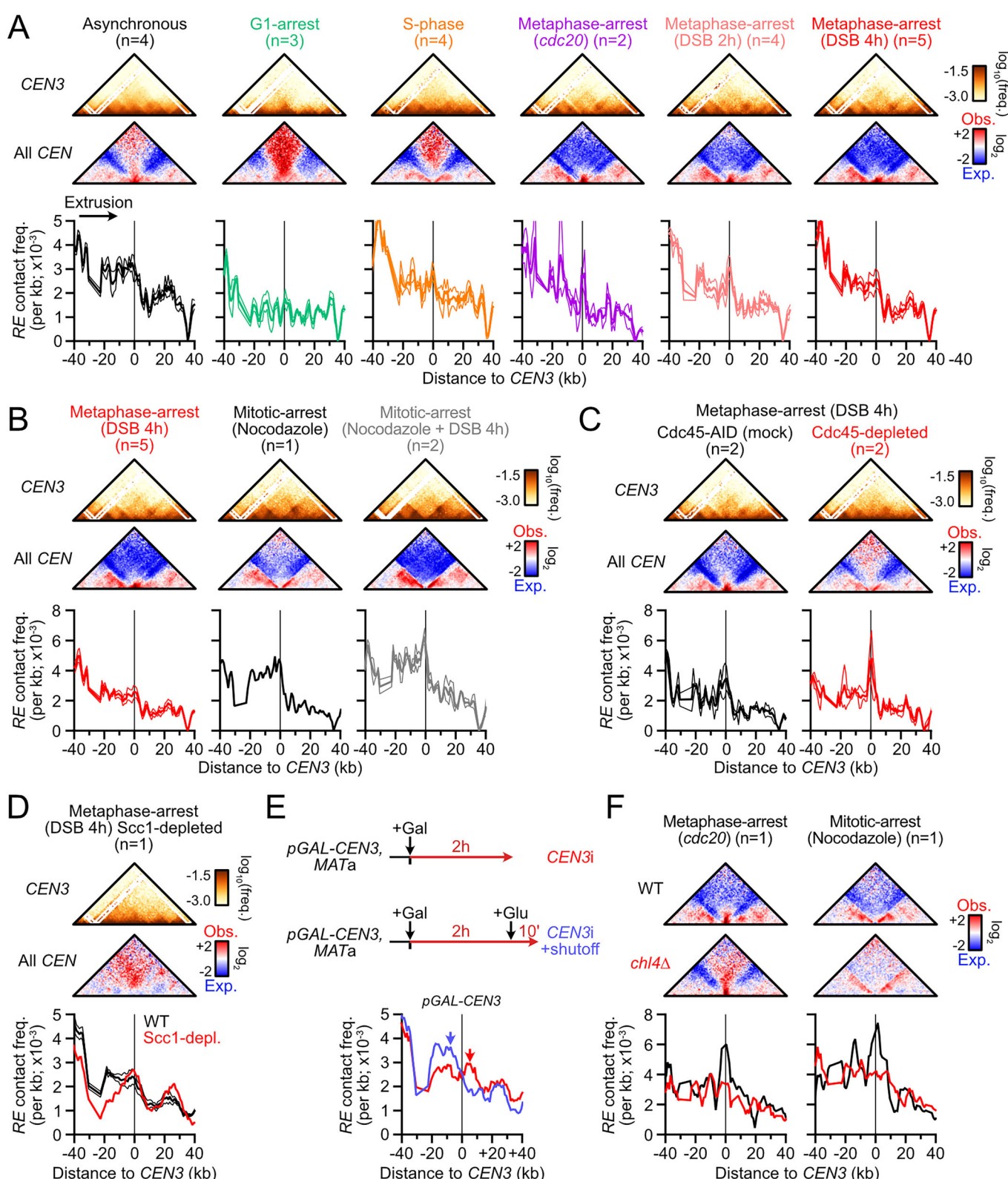

Dumont et al, 2024).

The efficiency of DSB induction, inferred from Hi-C coverage, was similar in all strains assayed (Fig. EV7A).

DSB induction resulted in the formation of a ~170 kb-long *RE-MAT* loop only in *MAT*a cells (Fig. 6A–C). Specifically, the contact stripe emanating from the *RE* was interrupted at the level of the DSB in the ~20 kb region immediately upstream of the DSB while contacts downstream were not different from control sites, indicating an absence of loop extrusion past the DSB (Fig. 6A–C,

**Figure 4.  Centromeres stall condensin translocation in a kinetochore-dependent manner.**

(A) Condensin roadblock at *CEN3* across the cell cycle. Top: Hi-C maps of the *CEN3*-surrounding region. Middle: Aggregated contact maps of all centromeres. Bin: 1 kb. Bottom: 4C-like profile centered on the *CEN3* region with the *RE* as a viewpoint. Data show mean ± SEM of zooms of Fig. 2A–E. *n* number of biological replicates. Data show mean ± SEM. (B) 4C-like profile with the *RE* as a viewpoint in cells arrested at mitosis in the presence or absence of nocodazole. *n* number of biological replicates. Data show mean ± SEM. (C) 4C-like profile with the *RE* as a viewpoint in metaphase-arrested cells with replicated chromatids (Cdc45-AID mock) or non-replicated chromatids (Cdc45-depleted prior to S-phase entry; APY513). Data show mean ± SEM of *n* = 2 biological replicates from (Dumont et al, 2024). (D) 4C-like profile with the *RE* as a viewpoint in metaphase-arrested cells with or without cohesin (Scc1-depleted, APY1481). Data show *n* = 1 biological replicate from (Dumont et al, 2024). (E) Top: *CEN3* inactivation scheme (APY1745). Bottom: 4C-like profile with the *RE* as a viewpoint in the *CEN3*-surrounding region. *n* = 1 biological replicate. (F) Aggregated contact maps of all centromeres (top) and 4C-like profile with the *RE* as a viewpoint (bottom) in WT and *chl4Δ* cells arrested in metaphase or in mitosis. Data show *n* = 1 biological replicate each from (Paldi et al, 2020).

and see below). A *MAT*a strain with an unrepairable DSB induced at *ura3* on chr. V did not exhibit such arrest, showing that the DSB exerts its roadblock effect in *cis*. Deletion of the Fkh1-containing left part of the *RE* did not affect formation of the DSB-induced *RE-MAT*a loop (Figs. 6D,E and EV7B,C), indicating that the condensin-containing right part of the *RE* is sufficient for loop formation. Inversion of this *RE-right* element abolished loop formation (Fig. 6D,E), establishing that the directional translocation of condensin towards the DSB is required to form the *RE-MAT*a loop. These observations show that a DSB at *MAT*a blocks the condensin-mediated loop extrusion initiated at the *RE* in *cis*, converting a heterogeneous population of loops along chr. III into a site-specific *RE*-DSB loop (Fig. 6F).

The *RE-MAT* loop resulted in broken *MAT*a engaging *HML*α ~3.2-fold more than *HMR*a (Fig. EV7D–F). It is opposite to the situation in *MAT*α cells, where *HMR*a is preferred ~twofold over *HML*α, leading to a ~6.4-fold difference for donor preference between both mating-types upon DSB formation (Fig. EV7F). This donor preference is only ~1.7-fold in the absence of a DSB (Fig. EV7D,F). Consequently, the structure that drives preferential interaction between *MAT*a and its target *HML*α donor is not the heterogeneous loop folding observed in asynchronous *MAT*a cells, but the *RE-MAT* loop specifically formed upon DSB formation. This site-specific loop is likely the relevant structure in promoting *MAT*a-to-α switching (see below, Fig. 6F and "Discussion") (Li et al, 2019; Dinda et al, 2023).

Intriguingly, a stripe emanating from the *RE* and stretching up to the DSB site appeared 4 h post-DSB induction in *MAT*α cells, bringing the DSB in close proximity to *HML*α (Fig. EV7G,H). This loop only formed after the elimination of the *MAT*α genes by resection (Fig. EV7A), whose α2 gene product represses the Mcm1-mediated condensin loading at the *RE* (Li et al, 2019; Dinda et al, 2023). Hence, the active repression of condensin loading at the *RE* in *MAT*α cells may provide a back-up redirection of homology search towards *HML*α in case of a prolonged failure to repair a *MAT*α DSB using the primary *HMR*a donor (Fig. EV7I).

## The *RE* and a DSB are sufficient to establish a directional loop

In order to address whether other elements on chr. III participate in *cis* in the establishment of the *RE*-DSB loop and to ascertain its formation in a Rad51-proficient background, we induced a single unrepairable DSB 165 kb downstream of the *RE-right* construct on chr. IV in *MAT*a and *MAT*α WT cells (Figs. 7A,B and EV8A; the endogenous HO cut-site at *MAT* was inactivated by a single point

mutation). This minimal system recapitulated the loop observed on chr. III (Fig. 7A–C). Inverting the *RE-right* construct abolished loop formation (Fig. 7A,B), demonstrating that the *RE*-DSB loop is formed as a consequence of condensin translocating towards the DSB in *cis*. Hence, formation of a *RE*-DSB loop reflects a basic property of condensin encountering a DSB that can operate outside of the natural context of chr. III and *MAT*, and in a Rad51-proficient context. The pause frequency of condensin at the DSB was estimated to be ~90% in this simple system (Fig. 7C), confirming the strong or absolute roadblock a DSB represents for condensin in cells. This roadblock did not affect the density of condensin on chromatin (compare Figs. 2G and 7C), suggesting that condensin-mediated loop dissociation remains the same whether condensin is arrested at a DSB or actively extruding.

## The *RE*-DSB loop promotes *RE*-proximal homology identification

To address the functional relevance of this condensin-dependent *RE*-DSB loop during recombinational DNA break repair, we quantified D-loops formed by the left DSB end at two competing intra-chromosomal donors positioned on each side of the DSB in our minimal experimental system on chr. IV (Fig. 7D). The donors were introduced at intergenic locations and at positions and distances from the DSB emulating that of *HML*α and *HMR*a relative to *MAT* on chr. III (~168 kb to the left and ~64 kb to the right of the DSB site introduced at position 845,464, respectively). The *RE-right* construct was further introduced near the left donor in the forward or inverted orientation, only the former of which leads to the formation of a *RE*-DSB loop (Fig. 7A,B). Absolute D-loop levels at each donor were quantified 1 and 2 h post-DSB induction using the D-loop-Capture assay (Piazza et al, 2019; Reitz et al, 2022; Djeghmoum and Piazza, 2025) (Figs. 7D and EV8B,C). D-loop formation was biased fourfold toward the most proximal right donor (Fig. 7D,E). Introduction of the *RE-right* near the left donor halved this preference by promoting D-loop formation at that donor, while having no effect on D-loops formed at the right donor (Figs. 7D,E and EV8D). No such stimulation was observed if condensin extruded loops away from the DSB (*RE-right*-inverted) or if loop extrusion from the *RE-right* site was prevented in *MAT*α cells (Fig. 7E). Hence, homology identification at the *RE*-proximal donor is stimulated only in the context in which a condensin-dependent *RE*-DSB loop can be formed. Condensin whose translocation has been blocked at a DSB thus promotes homology search around its loading site.

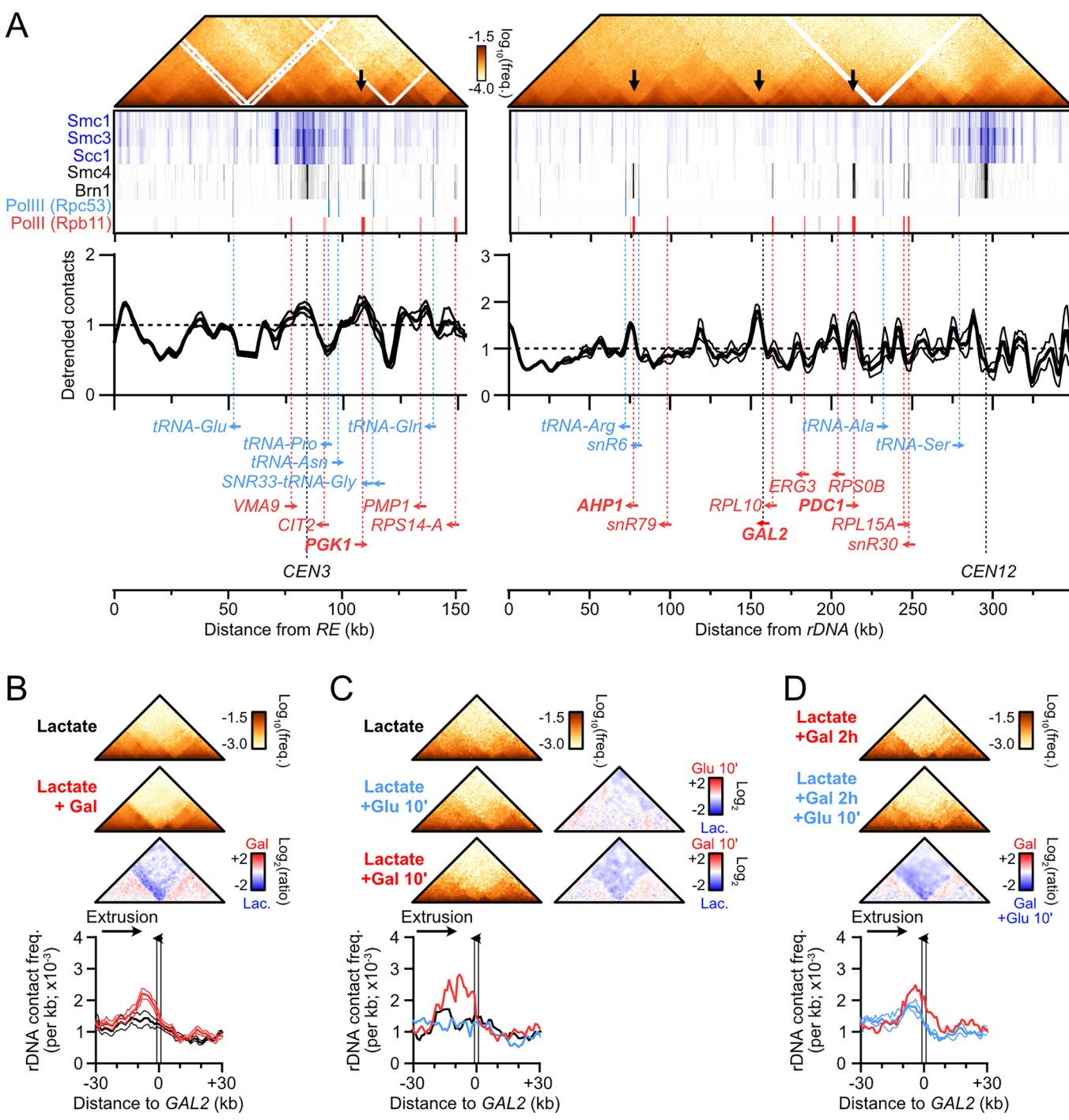

**Figure 5. Highly transcribed RNA PolII-dependent genes stall condensin translocation.**

(A) From top to bottom, Hi-C contact maps, ChIP-Exo enrichment profiles of cohesin, condensin, RNA PolIII, and RNA PolII subunits (from (Rossi et al, 2021)), detrended *RE* and rDNA-flanking 4C-like profiles in asynchronous *MAT*a cells (n = 4), and relevant genomic features. Contact data were obtained from cells grown in galactose-containing media, while ChIP-Exo data were obtained in glucose-containing media, explaining the discrepancy at *GAL2*. (B) Hi-C matrices (top) and 4C-like profiles with the left rDNA-flanking region as a viewpoint (bottom) in the 60 kb region surrounding the *GAL2* gene in asynchronous *MAT*a cells (APY266) grown in the presence of lactate or lactate supplemented with galactose. Data show mean ± SEM of n = 5 and 6 biological replicates, respectively. Bin: 1 kb. (C) Same as (B) in *MAT*a cells (APY142) grown in lactate media and supplemented or not with 2% galactose or 2% glucose for 10 min. Data show n = 1 biological replicate. (D) Same as (B) with samples used in Fig. 4E.

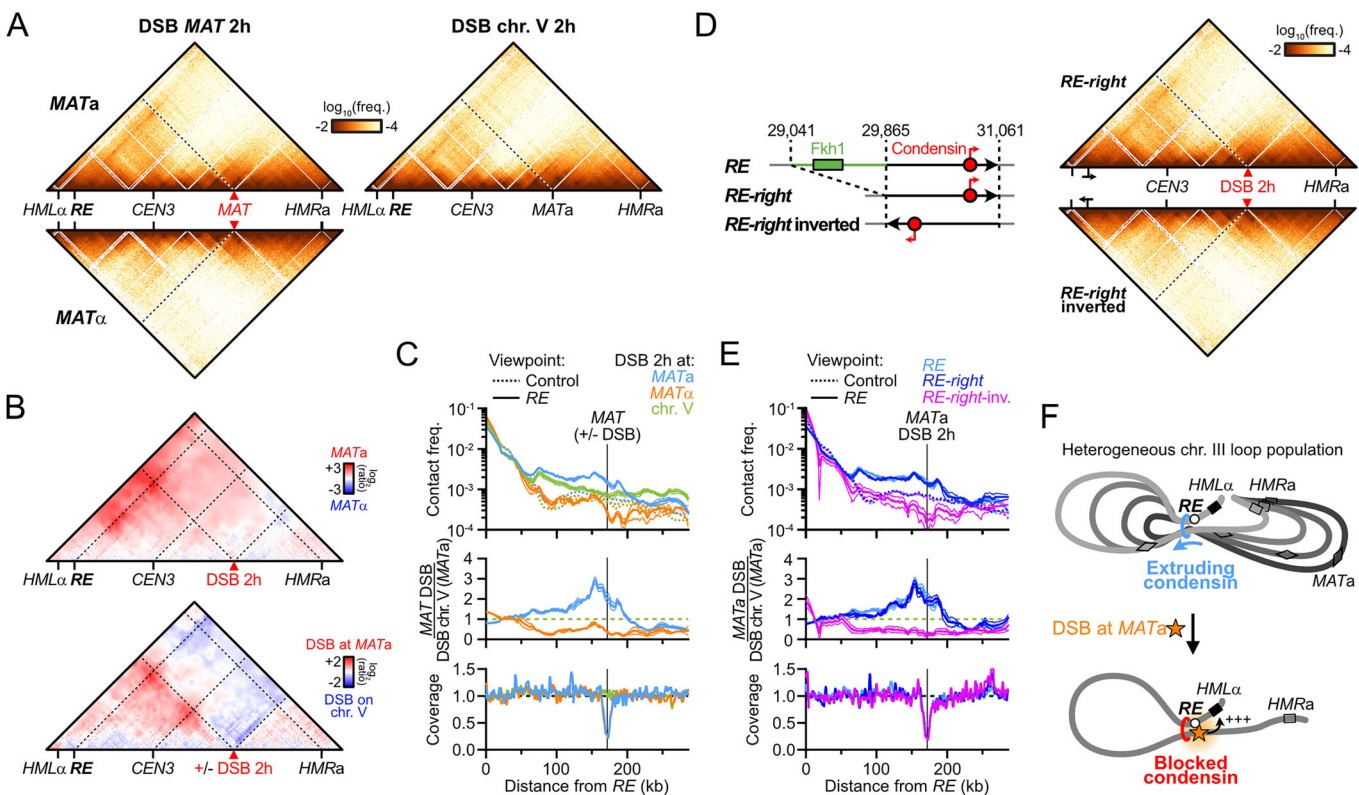

**Figure 6. A Rad51-independent *RE-MAT*a loop forms upon DSB induction.**

(A) Hi-C contact maps in *MAT*a and *MAT*α *rad51*Δ cells (APY1267 and APY1264, respectively) at 2 h post-DSB induction at *MAT* (n = 4 and 2 biological replicates, respectively). A *MAT*a strain with an unrepairable DSB on chr. V (APY266) is shown for comparison 2 h post-DSB induction (n = 4 biological replicates). Bin: 1 kb. (B) Ratio maps of data in (A). (C) Top: 4C-like profiles with the *RE* (or 6 control sites) as a viewpoint, from data in (A). Middle: Ratio of the 4C-profiles over that of a *MAT*a cell with a DSB on chr. V. Bottom: Coverage from Hi-C reads showing the resection tract length at *MAT* (see also Fig. EV7A). Data show mean ± SEM. (D) Left: Scheme of the *RE-right* and *RE-right*-inverted constructs. Right: Hi-C contact maps of *MAT*a *rad51*Δ cells with the *RE-right* and *RE-right*-inverted constructs (APY2072 and APY2079). Data show the merging of n = 3 biological replicates each. Bin: 1 kb. (E) Top: 4C-like profiles with the *RE*, *RE-right* and *RE-right*-inverted constructs (or 6 control sites) as a viewpoint, from data in (D). Middle: Ratio of the 4C-profiles over that of a *MAT*a cell with a DSB on chr. V. Bottom: Coverage from Hi-C reads showing the resection tract length at *MAT* (see also Fig. EV7B). Data show mean ± SEM. (F) Model for chr. III structure reconfiguration by condensin upon DSB formation into a homogeneous *RE-MAT*a loop. This structure juxtaposes the DSB and the *HML*α donor.

# Discussion

Here, we provided evidence that condensin extrudes chromatin loops from two specific sites in the budding yeast genome and determined several of its loop extrusion properties and roadblocks in cells. Notably, condensin forms a long-range loop structure between its loading site and a DSB, which promotes homology search at the loop anchor. We propose a revised model of donor selection during mating-type switching that exploits the specificities of loop extrusion by condensin we defined here.

## Properties of loop extrusion by condensin in *S. cerevisiae*

Contact stripes detected by Hi-C correspond to a heterogeneous population of loops sharing a discrete anchor. Such loops may be formed by a unidirectional extrusion process initiated at a loading site present at the base of the stripe (scenario 1), or by a site-specific block for unidirectional or bi-directional loop extrusion processes initiated at dispersed sites within the genomic interval covered by the stripe (scenario 2 and 3, respectively; Fig. 7F) (Fudenberg et al, 2016, 2017; Vian et al, 2018). Here, we present evidence in favor of

the first "loading" scenario. Indeed, in the "blocking" scenarios, the numerous ongoing loop extrusion events that have not yet reached the blocking site are expected to nonetheless link distant sites along the main Hi-C diagonal. Condensin removal should thus affect the probability of contact $P_c$ as a function of genomic distance $s$, which was not observed (Fig. EV1D). This absence of an effect on the genome-wide $P_c(s)$ could not be explained either by distributed loading events only on chr. III and the *CEN12-rDNA* intervals, as stripes could also be observed by introducing the *RE* on chr. IV (Fig. 2B). Importantly, the *RE*-DSB loops were only associated with a single stripe emanating from the *RE*, and not from the DSB (Figs. 6A,B and 7A,B). This observation is incompatible with scenarios 2 and 3, which both predict the appearance of stripes anchored at the DSB. These multiple lines of evidence indicate that condensin is loaded at the *RE*, from which it unidirectionally extrudes loops, as proposed in scenario 1 (Fig. 7D). This model is consistent with the unidirectional loop extrusion activity of condensin on naked DNA reported in vitro (Ganji et al, 2018; Kim et al, 2020; Shaltiel et al, 2022; Analikwu et al, 2025).

Intriguingly, loop extrusion by condensin from its loading sites has a defined orientation, leading to a single stripe, as observed

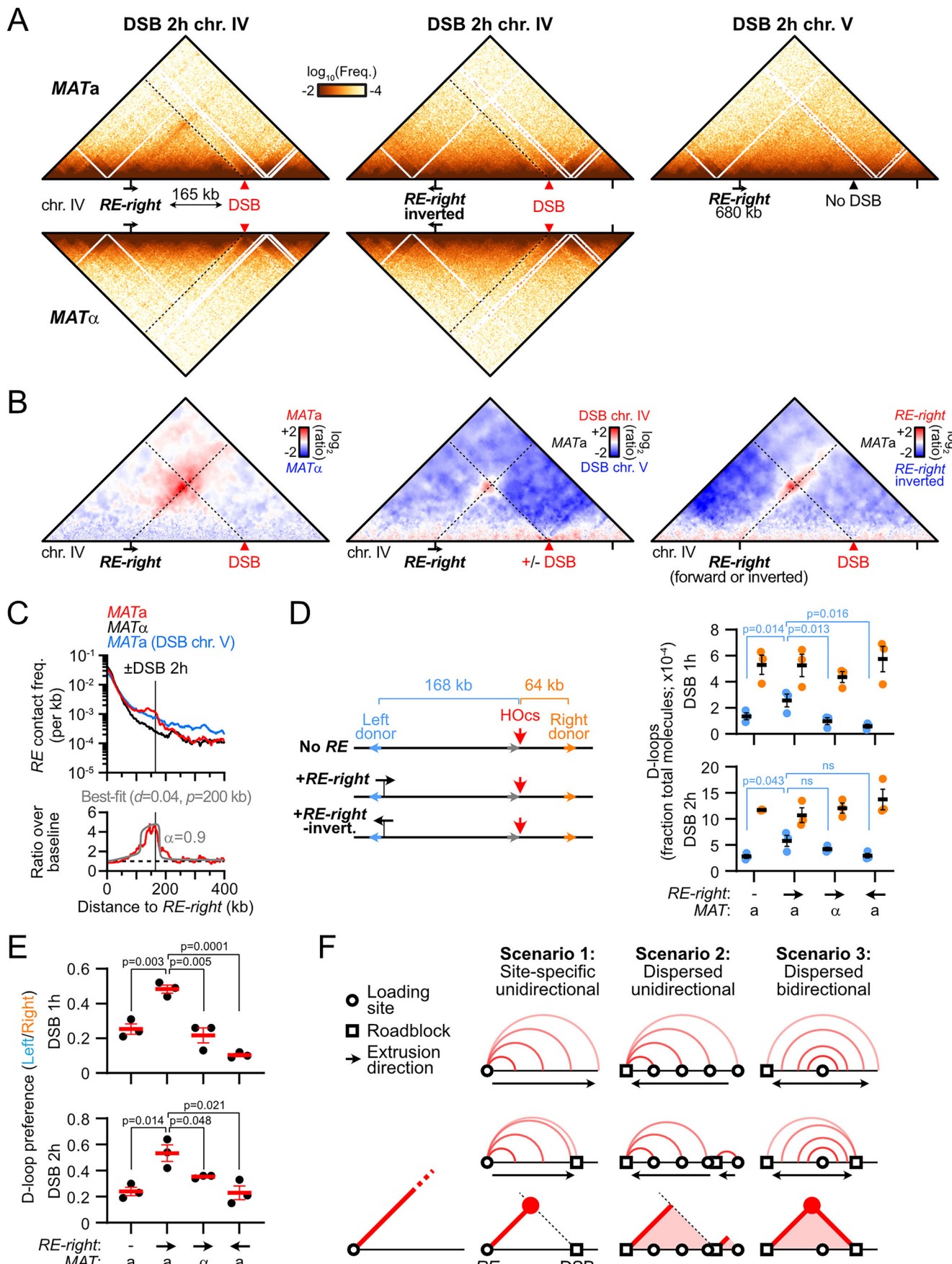

**Figure 7.  The *RE*-DSB loop is portable and promotes *RE*-proximal homology search.**

(A) Hi-C contact maps of the chr. IV region spanning from 555 to 970 kb that contains the *RE-right* constructs at position 680 kb and an unrepairable DSB at position 845 kb in *MAT*a (APY1850 and APY2058) and *MAT*α cells (APY1852 and APY2060), as well as in a *MAT*a strain with an unrepairable DSB on chr. V (APY1918). Data show $n = 1$ biological replicate. Bin: 2 kb. (B) Ratio maps of data in (D). (C) Top: 4C-like profiles from *RE-right* on chr. IV as a viewpoint, from data in (A). Bottom: Observed data and best-fit simulated contact frequencies. Data represent the ratio of *MAT*a over *MAT*α data. (D) Left: donor competition system to address the role of the *RE*-DSB loop in biasing homology search. Right: Absolute D-loop levels at the left and right donors at 1 and 2 h post-DSB induction in strains without (APY2083) or with the *RE-right* (*MAT*a: APY2085, *MAT*α: APY2087) or *RE-right*-inverted (APY2088) inserted near the left donor. Data show individual biological replicates ($n = 3$) and mean ± SEM. $P$ values were obtained using a paired ratio Student $t$ test. None of the comparisons for the right donor are significant. (E) Donor preference, computed from data in (D). Data show individual biological replicates ($n = 3$) and mean ± SEM. $P$ values were obtained using a Student $t$ test. (F) Predicted contact patterns with different loop extrusion scenarios in the presence of a DSB roadblock. The observed data in (B) correspond to scenario 1. Source data are available online for this figure.

upon introduction of the *RE* on chr. IV (Figs. 2B–D and EV2A,B). The mechanism specifying this orientation remains to be determined.

Condensin extrudes loops with a processivity ~150–250 kb, similar to that reported in a recent preprint using an orthogonal microscopy approach (preprint: Zou et al, 2025). Processivity does not substantially vary between the G1 and G2/M phases of the cell cycle, but is reduced during the S-phase. Condensin-mediated loop density is modestly decreased in G1, consistent with its overall reduced presence on chromatin and the lower amount of its limiting Ycg1 component at that cell cycle phase (D'Ambrosio et al, 2008; Leonard et al, 2015; Doughty et al, 2016). Finally, Top2 activity is required for condensin-mediated loop extrusion (Fig. EV4), similar to the X-specific condensin DC in *C. elegans* (Morao et al, 2022). It suggests a broad conservation of the coupling between the strand passage activity of Top2 and the loop extrusion activity of condensin, which likely underlies their shared function in chromatid condensation and decatenation during mitosis (Wood and Earnshaw, 1990; Hirano, 2016; Goloborodko et al, 2016; Racko et al, 2018; Orlandini et al, 2019; Dyson et al, 2021).

## Roadblocks for loop extrusion by condensin

Arrays of the high-affinity DNA-binding protein Rap1, naturally present at budding yeast telomeres, have been reported to block loop extrusion by condensin in vitro and in cells (Guérin et al, 2019; Analikwu et al, 2025). Here, we identified several additional natural roadblocks of varying intensity for condensin-mediated loop extrusion and delineated their main requirements:

- Centromeres are permeable roadblocks in a kinetochore-dependent, but microtubule-, cohesin-, sister chromatid- and thus tension-independent manner.
- RNA PolII genes are weak roadblocks, detected only at the most highly transcribed genes in both the G1 and G2/M phases of the cell cycle. These roadblocks can be induced and shut off within 10 min in *S. cerevisiae*, suggesting that RNA PolII activity itself, or its immediate consequences on chromatin composition and structure, hinders loop extrusion by condensin. The conservation of this roadblock in fission yeast (Lebreton et al, 2024) and *B. subtilis* (Gruber and Errington, 2009; Brandão et al, 2019) further hints at a basic interference between RNA polymerases and condensin activity, rather than species-specific management of highly transcribed regions (e.g., delocalization to the nuclear periphery).
- Replication forks are strong roadblocks for condensin, independent of their progression. Partial recovery of loop extrusion in S-phase

upon depletion of the origin-firing factor Cdc45 suggests that the fork structure itself is an obstacle for condensin translocation. This partial rescue also raises the possibility that specific replisome components recruited or remodeled upon S-phase entry could be obstacles outside of the context of forks, as proposed for cohesin and MCM complexes in mammalian cells (Dequeker et al, 2022). The inability of condensin to overcome replication forks and/or inactive replisome components has also been reported for the SMC-scpAB complex of *B. subtilis* (Liao et al, 2025). It suggests a conserved inability for loop-extruding condensin complexes to bypass replication forks, which may contribute to the condensation defects of unreplicated chromosomal regions in mammalian cells (Ono et al, 2013; Boteva et al, 2020).

- DNA double-strand breaks are absolute condensin roadblocks, independently of Rad51.

Whether these various impediments to condensin loop extrusion involve species-specific protein-protein interactions or whether they are steric in nature, as previously shown with heterologous high-affinity DNA-bound elements in *S. cerevisiae* and in vitro (Pradhan et al, 2022; Analikwu et al, 2025), remains to be determined.

## The relevant role of condensin in *MAT*a-to-α switching is to tether the DSB near *HML*α

Condensin loading at the *RE* has previously been shown to promote directional *MAT*a-to-α switching (Li et al, 2019; Dinda et al, 2023). This modest stimulatory effect was presumed to arise as a consequence of the heterogeneous "horseshoe" folding of chr. III, believed to reduce the average distance between the DSB and its target *HML*α donor in the context of a diffusive 3D search (Coïc et al, 2006a; Belton et al, 2015; Lassadi et al, 2015; Li et al, 2019; Dinda et al, 2023). Here, we reveal that condensin juxtaposes *MAT*a and the *RE*-surrounding region that includes the *HML*α donor specifically upon DSB formation (Fig. 6F), and that such loop promotes homology identification near the *RE* in our minimal *RE*-DSB system on chr. IV (Fig. 7D,E). We thus propose that the DSB-induced *RE*-*MAT*a loop is the relevant structure promoting a-to-α switching. In this model, the heterogenous loop population observed in the absence of a break represents a futile structure whose role is to poise chr. III for rapid establishment of the *RE-MAT* loop upon DSB formation.

Increased proximity between *MAT* and the *RE* has been previously observed cytologically within less than 1 h of DSB induction, but whether it resulted from the recombination process itself, or whether it preceded it, was not established (Simon et al,

2002; Bressan et al, 2004; Houston and Broach, 2006; Avşaroğlu et al, 2016). Here we show that this association takes place in the absence of Rad51 (Fig. 6A–C), indicating that it is not a consequence of homology search or of the Rad51 filament interaction with the Fkh1-containing part of the *RE* (Renkawitz et al, 2013; Dumont et al, 2024). The condensin-mediated clamping of the DSB and the *RE* is expected to limit their diffusion. Consistently, the diffusion coefficient of the chromatin flanking the *MAT* locus rapidly decreases following DSB induction (Saad et al, 2014), unlike DSBs formed on other chromosomes (Dion et al, 2012; Miné-Hattab and Rothstein, 2012). Functionally, such clamping should promote the oversampling of the *RE*-surrounding region and considerably accelerate *HMLα* identification. Accordingly, artificial tethering between *HMLα*- and *MATα*-adjacent sites could partially outcompete usage of *HMRa* (Simon et al, 2002; Kostriken and Wedeen, 2001), with a magnitude similar to that contributed by condensin in *HMLα* usage in *MATa* cells (Li et al, 2019; Dinda et al, 2023).

*MAT* switching thus provides a model to study the basic mechanism that establishes selective interactions between chromosomal segments. Biasing for *HMLα* usage in *MATa* cells depends on two main modules in the *RE*: a condensin-loading module whose deletion reduces donor bias by twofold (Li et al, 2019); and a Fkh1-binding module active only in G2/M whose deletion almost abolishes *HMLα* usage (Wu and Haber, 1996; Sun et al, 2002; Coïc et al, 2006b) (Fig. 2A). This major role of Fkh1 at the *RE* specifically depends on its FHA domain (Li et al, 2012) and can act in *trans* (Coïc et al, 2006b; Lee et al, 2016) by recruiting the Rad51-ssDNA filament (Dumont et al, 2024; Renkawitz et al, 2013). We thus propose that the more modest role of condensin in promoting *MATa*-to-α switching may be to accelerate the establishment of the first link between the *RE* and the broken *MATa* locus, whose maintenance would subsequently be handed over to Fkh1 and the Rad51-ssDNA filament. This two-step scenario for establishing specific long-range interactions in *cis* along chromosomes thus consists of an initial phase of moderate specificity modulated by intrinsic properties of a SMC complex and its roadblock followed by a maintenance phase dependent on the protein-protein and protein-DNA affinities of two (or more) cognate DNA-bound factors.

## Cohesin and condensin loop extrusion properties differently regulate homology search

All four structurally related SMC complexes (i.e., cohesin, condensin, Smc5/6, and Mre11-Rad50-Xrs2) have been implicated in specific or general aspects of recombinational DNA repair in budding yeast. Notably, cohesin regulates homology search during the repair of accidental DNA break in both budding yeast and human cells (Covo et al, 2010; Piazza et al, 2021; Dumont et al, 2024; Teloni et al, 2025; Marin-Gonzalez et al, 2025) while condensin promotes directional *MATa*-to-α switching (Li et al, 2019; Dinda et al, 2023) and, as we show here, homology identification near its loading site (Fig. 7D,E).

Mechanistically, cohesin in *S. cerevisiae* and RecN in *C. crescentus* (Piazza et al, 2021; Chimthanawala et al, 2022; Dumont et al, 2024) were proposed to endow the RecA/Rad51-ssDNA filament with the ability to access distant chromatin regions upon directional motion on chromatin in *cis*. To achieve such directional motion, SMCs must anchor at or near the RecA/Rad51-ssDNA filament and thread chromatin unidirectionally from that anchor. In this framework, SMC's processivity and roadblocks determine the scanning range. Accordingly, we previously showed that cohesin could promote the identification of a donor in *cis* as a function of its processivity, which could be expanded in a *pds5* mutant (Piazza et al, 2021). Consistently, the span of RAD51 chromatin enrichment around site-specific DSB sites can be modulated in opposite ways upon depletion of NIPBL$^{Scc2}$ and WAPL (Teloni et al, 2025), and the identification of distant ectopic donors in *cis* was reduced in the absence of NIPBL$^{Scc2}$ in human cells (Marin-Gonzalez et al, 2025), suggesting a broad conservation of this layer of regulation of homologous recombination imparted by cohesin.

Differently, condensin promotes a specific interaction between the region surrounding its loading site (i.e., the *RE*) and the break site located in its processivity range (Fig. 6F). Such conditional *RE*-DSB clamping is made uniquely possible by the properties of loop extrusion by condensin we defined here: its site-specific and oriented loading at the *RE*, its unidirectionality, and its inability to bypass a DSB. Hence, distinct SMC loop extrusion properties are exploited to promote different search strategies in *cis*: long-range scanning by cohesin, and focal search for condensin.

## Methods

**Reagents and tools table**

| Reagent/resource | Reference or source | Identifier or catalog number |
|---|---|---|
| **Experimental models** | | |
| W303 *RAD5*$^+$ *S. cerevisiae* strains | This study | Table EV1 |
| **Recombinant DNA** | | |
| trp1::GAL-HO | Pannunzio et al, 2008 | https://doi.org/10.1016/j.dnarep.2008.02.003 |
| RE-right (RE-left-deleted) | This study | Dataset EV1 |
| RE-right-inverted (RE-left-deleted) | This study | Dataset EV1 |
| chrIV-845464::LY-HOcs | This study | Dataset EV1 |
| ura3::LY-HOcs | Piazza et al, 2017 | Dataset EV1 |
| chrIV-680258::RE | This study | Dataset EV1 |
| chrIV-680258::RE-inverted | This study | Dataset EV1 |
| chrIV-680258::RE-right | Piazza et al, 2021 | Dataset EV1 |
| chrIV-680258::RE-right-inverted | This study | Dataset EV1 |
| chrIV-680258::L0.6_donor | This study | Dataset EV1 |
| chrIV-680258::L0.6_donor-RE-right | This study | Dataset EV1 |
| chrIV-680258::L0.6_donor-RE-right-inverted | This study | Dataset EV1 |
| chrIV-911574::TRP1-L0.6_donor | This study | Dataset EV1 |
| **Antibodies** | | |
| Mouse anti-Myc clone 9E11 | Invitrogen | Cat# MA116637 |
| mouse anti-GAPDH clone GA1R | Invitrogen | Cat# MA515738 |
| HRP-conjugated rabbit anti-mouse IgG | Invitrogen | Cat# A16160 |
| **Oligonucleotides and other sequence-based reagents** | | |
| PCR primers | This study | Table EV3 |
| **Chemicals, enzymes, and other reagents** | | |
| Sodium DL-lactate | Sigma-Aldrich | Cat# L1375 |

| Reagent/resource | Reference or source | Identifier or catalog number |
|---|---|---|
| D-Galactose | Carl Roth | Cat# 4987 |
| Indol Indole-3-Acetic Acid (IAA = Auxin) | Sigma-Aldrich | Cat# I3750 |
| Dimethyl Sulfoxide (DMSO) 100% | Sigma-Aldrich | Cat# D8418 |
| Trioxsalen | Sigma-Aldrich | Cat# T6137 |
| Zymolyaze 100 T | Carl Roth | Cat# 9329 |
| Formaldehyde | Sigma-Aldrich | Cat# F8775 |
| EGS (Ethylene glycol bis succinic acid) | Fisher Scientific | Cat# 10350924 |
| Arima Hi-C kit | Arima Genomics | Cat# |
| Dynabeads™ Streptavidin C1 | Fisher Scientific | Cat# 10099482 |
| AMPure XP beads | Beckman Coulter | Cat# A63881 |
| Qubit DNA high-sensitivity kit | Thermofisher Scientific | Cat# Q32851 |
| EcoRI-HF | NEB | Cat# R3101 |
| T4 DNA ligase | NEB | Cat# M0202 |
| Proteinase K | NEB | Cat# P8107S |
| RNAseA | EUROMEDEX | Cat# 9707-C |
| SDS 20% | Fisher Scientific | Cat# 10607633 |
| Triton™ X-100 | Fisher Scientific | Cat# 10671652 |
| SSO Advanced Universal SYBR supermix | Bio-Rad | Cat# 1725274 |
| **Software** | | |
| Prism 10 | Graphpad | https://www.graphpad.com/ |
| CFX Maestro 2.0 | Bio-Rad | |
| Jupyter notebook 6.5.7 | JupyterLab | https://jupyter.org/ |
| **Other** | | |
| CFX96 Touch Deep Well Real-Time PCR Detection System | Bio-Rad | Cat# 3600037 |

## Haploid *S. cerevisiae* strains

Genotypes of *Saccharomyces cerevisiae* (W303 *RAD5+* background) strains are listed in Table EV1. The inducible *HO* expression construct *trp1::pGAL1-HO::hphMX*, the mutagenesis of the endogenous HO cut-site at *MAT*, and the DSB-inducible HOcs construct *LY-HOcs* at *ura3* on chr. V and at position 845,464 on chr. IV have been described previously (Piazza et al, 2019, 2018, 2021). The *rad51::KanMX* mutation has been obtained by transformation of a PCR fragment amplified from the relevant deletant of the Euroscarf gene deletion collection. The construct *his3::pADH1-OsTIR1-9Myc::HIS3* for OsTir1 E3-ubiquitin ligase expression, as well as the Scc1-V5-AID and Cdc45-FlagX5-AID constructs, have been described previously (Piazza et al, 2019; Dauban et al, 2020). The *SMC2-AID-9Myc::NAT* and *TIR1-9Myc::URA3* constructs were described previously (Guérin et al, 2019). The *pGAL1-CEN2* and *pGAL1-CEN3* constructs have been described previously (Hill and Bloom, 1987; Reid et al, 2008).

The extended recombination enhancer (*RE*) on chr. III (coordinates 29,041–31,071) or its condensin peak-containing right

segment (coordinates 29,865–31,071) have been introduced at an intergenic position on chr. IV (coordinate 680,258) upon CRISPR/Cas9-mediated knock-in using guide 5'-TTGTTTCTAC-TAATGTGCTG-3' and repair gene fragments bearing ~150 bp of homology to each side of the break as described in (Agier et al, 2021). The *RE* deletion on chr. III has been obtained by CRISPR/Cas9 as described in (Dumont et al, 2024). The deletion of the Fkh1-binding *RE-left* region (coordinates 29,041-29,865) has been generated upon CRISPR/Cas9-mediated targeting using two guides (5'-TCTCAAAACCAAATTGCGCA-3' and 5'-CCAATTC-CAAATTCTAGGGA-3') and a repair fragment bearing ~150 bp of homology to both sides of the desired deletion junction. The inversion of the condensin-binding *RE-right* region (coordinates 29,866–31,061) has been generated upon CRISPR/Cas9-mediated targeting using two guides (5'-TCTCAAAACCAAATTGCGCA-3' and 5'-TTGGCTCTATAAAGGAGTTC-3') and two repair fragments bearing ~150 bp of homology to both sides of the desired inversion junctions.

Two 556 bp-long donors (corresponding to the position +23 to +578 of the *LYS2* gene) were introduced at positions 680,088 and 911,499 on chr. IV upon CRISPR/Cas9-mediated targeting using two guides (5'-CAAGATACAAGCCGTTTCCA-3' and 5'-CGCAATGATGCAATAGTCCA-3') and two repair fragments containing the donor sequence bearing ~150 bp of homology to both sides of the desired insertion points. The donor at position 911,499 is flanked by a *TRP1* marker. They bear homology to the left end side of the DSB region introduced at position 845,464 on the same chromosome.

All the coordinates correspond to the S288c R64-2-1 *S. cerevisiae* genome assembly. All the genetic constructs generated in this study are available as annotated Genbank files in Dataset EV1.

## Culture media and growth conditions

### G1-arrest
Exponentially growing cultures in YEP-lactate-galactose (1% yeast extract, 1% peptone, 2% lactate, 2% galactose) were synchronized in G1 by the addition of 1 μg/ml alpha-factor (GeneCust) every 30 min for 3 h at 30 °C prior to crosslinking.

### S-phase
Cells arrested in G1 with alpha-factor in YPD medium (1% yeast extract, 1% peptone, 2% glucose) at 30 °C were washed 3 times with 50 mL of YPD and released in S-phase at 25 °C for 20 min prior to crosslinking. For Scc1-AID depletion and Cdc45-AID depletion, 2 mM IAA was added 1 h prior to, and in all following wash and culture media (described and controlled for by Western blot and FACS in (D'Asaro et al, 2025)).

### Metaphase-arrest (CDC20 repression)
Metaphase-arrest in strains in which the *CDC20* gene is placed under the control of the *pMET3* promoter (APY537) was performed as in (Dauban et al, 2020) with minor media differences. Exponentially growing cells in a supplemented synthetic complete lactate media deprived of methionine (0.67% yeast nitrogen base without amino acids, 2% lactate, and supplemented with a mix of amino acids lacking methionine) were arrested in G1 upon addition of 1 μg/ml alpha-factor (GeneCust) every 30 min for 4 h at 30 °C.

Cells were washed three times with 50 mL of YEP-lactate supplemented with 2 mM methionine, and maintained arrested in this media at 30 °C for 4 h prior to crosslinking.

### Metaphase-arrest (DNA damage checkpoint-induced)

Exponentially growing cultures in YEP-lactate medium of strains bearing the galactose-inducible *pGAL1-HO* construct for expression of the HO endonuclease and its unrepairable *HOcs* target site at *ura3* (APY266) were synchronized in G1 by addition of 1 µg/ml alpha-factor every 30 min for 3 h at 30 °C. Cells were washed three times with 50 mL of pre-warmed YEP-lactate and released in S-phase in YEP-lactate supplemented with 2% galactose to induce HO expression from the *pGAL1-HO* construct, which targets an unrepairable HOcs at *ura3* on chr. V (Piazza et al, 2018, 2019). In one instance, 10 µg/mL nocodazole was added to cause microtubule depolymerization. Cdc45-AID depletion was induced upon addition of 2 mM IAA 1 h prior to release in S-phase and maintained in all media thereafter. Scc1-AID depletion was induced upon addition of 2 mM IAA upon release in S-phase and HO induction (described and controlled for by Western blot and FACS in (Dumont et al, 2024)). Cells were crosslinked at 2 and 4 h post-DSB induction.

### Mitotic-arrest (DNA damage checkpoint-induced)

Exponentially growing cultures in YEP-lactate medium at 30 °C were supplemented with 10 µg/mL nocodazole and cultured for an additional 4 h prior to crosslinking.

### Induction of a DNA double-strand break at MAT

Exponentially growing Rad51-deficient cells (APY1264 and APY1267) in YEP-lactate medium at 30 °C were supplemented with 2% galactose to induce the expression of the HO endonuclease from the *pGAL1-HO* construct. Cells were crosslinked at 2 and 4 h post-DSB induction.

## Flow cytometry

Approximately $10^7$ cells were collected by centrifugation, resuspended in 70% ethanol, and fixed at 4 °C for at least 24 h. Cells were pelleted, resuspended in 1 mL of 50 mM sodium citrate pH 7.0, and sonicated for 10 s on a Bioruptor. After washing, cells were treated with 200 µg of RNase A (Euromedex, cat. 9707-C) at 37 °C overnight. Cells were then washed and incubated for 30 min with 1 mL of 50 mM sodium citrate pH 7.0 with 16 µg of propidium iodide (Fisher Scientific, 11425392). Flow cytometry profiles were obtained on a MACSQuant machine and analyzed using Flowing Software 2.5.1.

## Protein extraction and western blotting

Protein extracts for western blot were prepared from $5.10^7$ to $10^8$ cells. Cells were lysed in cold NaOH buffer (1.85 N NaOH, 7.5% v/v beta-mercaptoethanol) for 10 min in ice. Proteins were precipitated upon addition of trichloroacetic acid (15% final) for 10 min in ice. After centrifugation at $15,000 \times g$ for 5 min, the pellets were resuspended in 100 µL of SB + + buffer (180 mM Tris-HCl pH 6.8, 6.7 M urea, 4.2% SDS, 80 µM EDTA, 1.5% v/v beta-mercaptoethanol, 12.5 µM bromophenol blue). Proteins were denatured upon heating 5 min at 65 °C. Pre-cleared extracts were

resolved on 12% precast polyacrylamide gel (Bio-Rad, cat. 4561043) and blotted on a PVDF membrane (GE Healthcare, cat. 10600023). Membranes were probed with mouse anti-Myc monoclonal antibody (clone 9E11, Invitrogen, cat. MA116637) diluted at 1:1000 for Smc2-Myc-AID, and a mouse anti-GAPDH monoclonal antibody (clone GA1R, Invitrogen, MA515738) diluted at 1:5000. Primary antibodies were revealed with an HRP-conjugated rabbit anti-mouse IgG antibody diluted at 1:5000 (Invitrogen, A16160) using Immobilon Forte western HRP substrate (Merck, WBLUF0100) and a Chemidoc MP Imaging system (Bio-Rad).

## Hi-C

Hi-C was conducted as described in (Dumont et al, 2024) with minor modifications. Briefly, $\sim 1.5 \times 10^9$ haploid cells were fixed with 3% formaldehyde (Sigma-Aldrich, cat. F8775) for 30 min at RT at with orbital agitation at 120 rpm. Formaldehyde was quenched with 330 mM glycine for 20 min at RT at 120 rpm. Cells were washed twice with cold water at $3000 \times g$ for 10 min. Pellets were split into two tubes and frozen at -80 °C. The crosslinked pellets were thawed in ice and resuspended in cold $H_2O$ supplemented with an anti-protease mix (Roche, cat. 11836170001). Approximately $7.5 \times 10^8$ cells were transferred to a Precellys VK05 tube, and lysed for $3 \times 30$s at 6,800 rpm. Between 2 and $5.10^7$ cells were processed for Hi-C using the Arima Hi-C+ kit (Arima Genomics, cat. A410079) following the manufacturer's instructions. The Arima Hi-C+ kit employs a dual restriction digestion (DpnII and HinfI) yielding a median fragment length of 108 bp in *S. cerevisiae*.

DNA was fragmented into 300-400 bp fragments using the Covaris M220 sonicator. Preparation of the libraries for paired-end sequencing on an Illumina platform was performed using the Thermofisher Collibri ES DNA Library Prep Kit for Illumina Systems with UD indexes (cat. A38606024) following the manufacturer's instructions. The library was amplified in triplicate PCR reactions using oligonucleotides corresponding to the Illumina sequence adapters (5'-AATGATACGGCGACCACCGAGATCTA-CAC-3' and 5'-CAAGCAGAAGACGGCATACGAGAT-3') and Phusion DNA polymerase (New England Biolabs, cat. M0531) for 11 cycles. PCR products were purified with AMPure XP beads (Beckman Coulter, cat. A63881) and resuspended in pure $H_2O$. The Hi-C library is quantified using the Qubit DNA high-sensitivity kit (Thermo Scientific, cat. Q32851) on a Qubit 2 fluorometer (Thermo Scientific, cat. Q32866). Library quality control, paired-end sequencing ($2 \times 150$ bp) on Illumina NovaSeq6000 or NovaSeq X Plus, and data QC were performed by Novogene UK. The correspondence between Hi-C libraries and figure panels are listed in Table EV2.

## D-loop capture

D-loop Capture has been performed as described in (Reitz et al, 2022; Djeghmoum and Piazza, 2025). The rationale is depicted in Fig. EV8B. Briefly, cells were collected at 4 °C and resuspended in a solution containing 0.1 mg/mL Trioxsalen and crosslinked upon 365 nm UV irradiation on a BIO-LINK irradiator (Vilber-Lourmat, Cat. BLX-365) for 10 min. Cells were spheroplasted and lysed in the presence of 4 pM of APO563 (i.e., an oligonucleotide enabling

restoration of a restriction site on the resected broken molecule, Table EV3. The DNA was digested with EcoRI-HF (NEB, cat. R3101L) and ligated in dilute conditions ($\simeq 1.8 \times 10^4$ genome/μL) with T4 DNA ligase (NEB, cat. M0202). DNA was extracted with phenol-chloroform following protein digestion with proteinase K. Psoralen inter-strand crosslinks and adducts were reversed in 100 mM KOH at 90 °C for 30 min. The pH was neutralized upon addition of 66 mM of NaoAc pH 5.2. Approximately $6 \times 10^5$ genome equivalent were used per quantitative PCR (qPCR) reaction, performed in duplicate, on a CFX96 Touch Deep Well Real-Time PCR Detection System (Bio-Rad cat. 3600037), using the SsoAdvanced Universal SYBR Green Supermix (Bio-Rad, cat. 1725274), following the manufacturer's instructions. Primers used are listed in Table EV3. qPCR analysis was performed as described in (Reitz et al, 2022) using Bio-Rad CFX Maestro and Microsoft Excel.

## Data retrieval from SRA and GEO

Raw paired-end reads from Hi-C experiments in PRJNA526833 (Paldi et al, 2020), PRJNA680815 (Jeppsson et al, 2022), and PRJNA986466 (Jeppsson et al, 2024) were retrieved using the nf-core/fetchngs pipeline. ChIP-Exo profiles from (Rossi et al, 2021) (GEO GSE147927 series) were obtained from https://www.datacommons.psu.edu/download/eberly/pughlab/yeast-epigenome-project/.

## Hi-C reads alignment and generation of contact maps

The Hi-C reads alignment and the generation of contact maps were achieved using the Reads_to_hic.sh script. Briefly, paired-end 150 bp-long reads in fastq.gz format were pre-digested with DpnII and HinfI using parasplit in "all-vs-all" mode. Alternatively, reads were digested using the "cutsite" mode of the Hicstuff *pipeline* function. Hicstuff *pipeline* was used to align pairs of reads independently using Bowtie2 and generate contact data in Graal format (Matthey-Doret et al, 2022) using a modified *S. cerevisiae* R64-2-1 reference genome in which the *HML, MAT, HMR, URA3, LYS2*, and the right rDNA repeat loci were replaced by "N" of the same length (S288c_DSB_chr3_rDNA reference). In instances where a second *RE* fragment was introduced on chr. IV, the *RE* sequence on chr. III was also masked (S288c_DSB_chr3_rDNA_RE_N reference). Each uniquely mapped read was assigned to a restriction fragment; the uncut, circularization, and spurious ligation events were filtered out as described in ref. (Cournac et al, 2012) and PCR duplicates removed. The argument -d was set to compute the per-chromosome probability of contact $P_c$ as a function of the genomic distance $s$ from pairs files. The resulting sparse matrix in Graal format was binned at 1 kb resolution using the Hicstuff *rebin* function and converted to a cooler format with the Hicstuff *convert* function. ICE normalization of the cooler file was performed using the Cooler *balance* function (Abdennur and Mirny, 2020). Iterative coarsening was achieved using the Cooler *zoomify* function, which resulted in a mcool file. Graal and cooler files were used for downstream analysis.

## Generation of Hi-C maps

Hi-C maps were generated from sparse Hi-C data in Graal or cooler format using Hicstuff *view* function, balanced using the SCN method (Cournac et al, 2012), log-transformed, and binned with the Plot_matrix_using_hicstuff.sh script. Alternatively, ratio maps were generated from cooler files using Serpentine (Baudry et al, 2020) using the Ratio_map_Serpentine_from_cooler.ipynb script. Briefly, two matrices of the region of interest binned at 1 kb were subsampled to contain the same number of contacts and converted to a dense format. Comparison of contact maps was performed using the default threshold parameters (50 and 5) following 30 serpentine binning iterations. The trend was set to 0.

## Generation of aggregated Hi-C contact maps

Intra-chromosomal centromere and *GAL* genes pile-ups were generated with Chromosight *quantify* with default parameters (Matthey-Doret et al, 2020) from ICE-normalized Hi-C matrices in cooler format binned at 1 kb and subsampled at 24 million contacts using the Chromosight_pileup_GALgenes.sh and Chromosight_pileup_centromeres.sh scripts. The pile-ups show the contact enrichment at the coordinates of interest over random genomic sites.

## Computation of the contact probability as a function of genomic distance

Computation of the contact probability as a function of genomic distance $P_c(s)$ and its derivative has been determined from the per-chromosome file generated by Hicstuff *pipeline*. The contact decay probability of the mitochondrial genome and the endogenous 2μ plasmid were removed using the DistanceLaw1_remove_mito_and_2_micron.sh script, and the genome average $P_c(s)$ and slope were computed using the *distance law* function of the Hicstuff package with default parameters within a reference window of 3 and 300 kb using the DistanceLaw2_compute_genome-wide_average.ipynb notebook.

## Quantification of coverage from Hi-C data

Quantification of coverage from *bam* alignment files generated with the Hicstuff *pipeline* function were sorted and merged with Samtools *sort* and *merged* function, respectively, as part of the Reads_to_hic.sh pipeline. The coverage was computed in non-overlapping 500 bp bins using Tinycov *covplot* function. The coverage was normalized onto the median genome coverage, and divided over that of metaphase-arrested cells lacking a DSB on chr. III or chr. IV for computing resection tracts, or on G1-arrested cells for mapping replication progression in S-phase. These alignment tracts were also used to confirm the presence of the expected genetic constructs in each strain.

## Quantification of coverage from ChIP-Exo data

Quantification of coverage from *bam* alignment files following sorting with samtools *sort* function were computed as non-overlapping 20 bp bins using Tinycov *covplot* function. The resulting bedgraph files were rendered as histograms or heatmaps on IGV.

## Extraction of intra-chromosomal 4C-like contact profiles

Intrachromosomal 4C-like contact profiles were generated using the 4C_like_condensin_ICE.ipynb script as described in Fig. 1B.

Sparse Hi-C matrices binned at 1 kb in cooler format were loaded, ICE-normalized, and converted to dense format using the *flexible_loader* and *sparse_to_dense* function of the Hicstuff package, respectively. Vectors corresponding to the intra-chromosomal contacts made by the 3 kb *RE*-containing region to the right end of chr. III, the 3 kb region flanking the rDNA on the centromere-proximal side toward the left end of chr. XII, and/or the 3 kb region surrounding the insertion point for the *RE* on chr. IV:680,258 were extracted, smoothed with the Savitzky-Golay filter using the *signal.savgol* function of Scipy, and the sum of contacts set to 1. The smoothing kernel size was set to 39 kb with a polynomial order of 2 for whole chromosomes, and to 9 kb with a polynomial order of 1 for zoom on specific features, except otherwise stated. For each viewpoint, 6 control viewpoints located at the same distance from a centromere and on chromosome arms of a similar length were selected, processed in the same way, and their contact frequency averaged. Control viewpoints for the *RE*, in the form chr:coordinate(strand), were V:66000(W), VIII:21000(W), XIV:713000(C), II:153000(W), XI:525631(C), XIII:182646(W). Control viewpoints for the left rDNA-flanking region were IV:747000(C), XIV:332000(W), XIII:565000(C), XV:624000(C), XI:143000(W), II:536000(C). Data (mean ± SEM) of traces corresponding to independent biological replicates are plotted using Graphpad Prism as a distance to the viewpoint. Ratio between two conditions was manually computed under Microsoft Excel.

Computation of the 4C-like contact profiles with the left side of *MAT* as a viewpoint (III:190000) was computed on Hi-C matrices binned at 5 kb. No smoothing was applied.

## Statistical analysis

Statistical tests and linear regressions were performed under Graphpad Prism 10. The experiments were not randomized. The investigators were not blinded to allocation during experiments and outcome assessment. The number of times an experiment has been repeated ($n$), the nature of the data representation (mean, SEM, etc.), and the statistical test used are indicated in the figure panels or legends. A Student $t$ test was used to compare DLC data. Donor preference computed the ratio of D-loops formed at each donor within a single sample. These distributions were compared using an unpaired two-tailed Student $t$ test. The absolute DLC data were compared using a paired ratio Student $t$ test. This test was chosen because of batch-level variations in D-loop retrieval, which originates from subtle day-to-day differences in crosslinking efficiency (Reitz et al, 2022).

## Loop extrusion model

We modeled extrusion as a unidimensional process working along the chromatin: (1) condensin can bind and unbind at a single loading site at rates $k_b$ and $k_u$, respectively; (2) upon binding, one leg of the condensin remains anchored at the binding site while the other translocates along the chromosome at a speed $v_e$. In the idealized scenario of continuous processing without boundaries or roadblocks, the distribution of loop sizes extruded by condensin, $l$, follows an exponential distribution (Brackley et al,

2017; Abdulla et al, 2022):

$$\rho_0(l) = \frac{1}{p} e^{-l/p} \tag{1}$$

where $p = v_e/k_u$ is the processivity and represents the average loop size extruded by condensin when it is bound to chromatin. When an impermeable roadblock is present at site $i$, located a genomic distance $l_i$ from the loading site, the loop size distribution is modified as follows:

$$\rho_i(l, l_i) = \begin{array}{ll} e^{-\frac{l}{p}}/p & l < l_i \\ e^{-l_i/p} & l = l_i \\ 0 & l > l_i \end{array} \tag{2}$$

Assuming a density $\alpha_i$ of roadblocks at block site $i$, the total loop size distribution for $n$ discrete roadblocks is given by

$$\rho(l) = \left[\prod_{k=1}^{n}(1-\alpha_k)\right]\rho_0 + \sum_{i=1}^{n}\left[\prod_{k=1}^{i-1}(1-\alpha_k)\alpha_i\right]\rho_i(l, l_i). \tag{3}$$

If roadblock sites are not discrete but rather domains of width $\{\Delta l_{ki}\}$, we assume that one blocking barrier would be positioned randomly within the domain. Therefore, there are $\prod_{k=1}^{n}\Delta l_k$ different possible configurations of block site positions. To each possible configuration $\{l_i\}$ corresponds a size distribution (via Eq. (3)). The total loop size distribution is thus the mean value of $\rho(l)$ averaged over all possible block site configurations.

Given a loop size distribution, we can compute the 4 C contact profile between the loading site and other genomic positions using a Gaussian polymer approximation. The contact frequency in the absence of loop extruders is described by (Halverson et al, 2014):

$$P_0(s) = As^{-\gamma}, \tag{4}$$

where $A$ is a constant, $s$ is the genomic distance from the loading site, and $\gamma$ is 1.5 for an ideal polymer and less for more compact regions (Mirny, 2011). In the presence of a fixed loop of size $l$, the contact probability is (Polovnikov et al, 2023; Polovnikov and Slavov, 2023):

$$P(s, l) = \begin{array}{ll} A[s(l-s)/l]^{-\gamma} & s < l \\ 1 & s = l \\ A[s-l]^{-\gamma} & s > l \end{array} \tag{5}$$

For simplicity, we assume $A = 1$. The total 4 C profile is then given by:

$$P(s) = (1-d)P_0 + d\sum_l \rho(l)P(s, l) \tag{6}$$

with $d = k_b/(k_b + k_u)$ the probability to have a condensin bound at the loading site and extruding a chromatin loop.

By adjusting the parameters of the model ($p, d, \{\alpha_i\}, \{l_i\}, \{\Delta l_i\}$), we aim to achieve the best-fit with experimental observations. In summary, we compare the ratio of experimental contact frequency to its corresponding control with the theoretical prediction of $P(s)/P_0(s)$. By iteratively adjusting the model parameters, we identify the optimal values. Additionally, we estimate the parameters associated with potential roadblocks by analyzing the heights, lengths, and positions of significant peaks located near these roadblocks.

## Software used

- nf-core/fetchngs (version 1.12.0 available at https://github.com/nf-core/fetchngs)
- Hicstuff (version 3.2.4 available at https://github.com/baudrly/hicstuff).
- Chromosight (Matthey-Doret et al, 2020) (version 1.6.3, available at https://github.com/koszullab/chromosight).
- Serpentine (Baudry et al, 2020) (version 0.1.3, available at https://github.com/koszullab/serpentine).
- Tinycov (version 0.3.1, available at https://github.com/cmdoret/tinycov).
- Integrative Genomics Viewer (Robinson et al, 2011) (version 2.8.3, available at https://igv.org).
- Bowtie2 (Langmead and Salzberg, 2012) (version 2.3.5.1 available online at http://bowtie-bio.sourceforge.net/bowtie2/).
- Samtools (Danecek et al, 2021) (version 1.3.1 available online at https://github.com/samtools/samtools).
- Cooler (Abdennur and Mirny, 2020) (version 0.9.1 available online at https://github.com/open2c/cooler).
- Parasplit (version 1.1.5 available at https://gitbio.ens-lyon.fr/LBMC/hub/parasplit/).
- Flowing Software (version 2.5.1 freely available online at https://flowingsoftware.com/download/).
- Graphpad Prism (version 10, commercially available at https://www.graphpad.com/).

## Data availability

The newly generated raw Illumina paired-end sequencing data have been deposited to the NCBI Sequence Reads Archive (SRA) under accession PRJNA1169376. HiC data in cooler format binned at a 1 kb resolution have been deposited to NCBI Gene Expression Omnibus (GEO) under accession GSE278899. The correspondence between figure panels and Hi-C libraries is provided in Table EV2. The loop extrusion model is available as a Jupyter Python notebook at https://github.com/physical-biology-of-chromatin/CondensinYeast. Scripts, input data and reference genomes used for Hi-C data analysis are available at https://github.com/Piazzalab/Piveteau_condensin_2026.

The source data of this paper are collected in the following database record: biostudies:S-SCDT-10_1038-S44318-026-00748-6.

## Peer review information

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

## Acknowledgements

We thank members of the Piazza, Bernard, and Jost laboratories, as well as Armelle Lengronne, Frédéric Beckouët, and Romain Koszul for helpful discussions. We are grateful to Pascal Bernard, Armelle Lengronne, Jim Haber and Stéphane Marcand for their critical reading of the manuscript, and Rodney Rothstein, Thomas Guérin and Stéphane Marcand for sharing yeast strains. This research was supported by the European Research Council (ERC) under the European Union's Horizon 2020 (ERC grant agreement 851006), and the Fondation ARC pour la Recherche sur le Cancer www.fondation-arc.org (ARCPGA2022110005583_6379) to AP, and the Agence Nationale de la Recherche to AP and DJ (ANR-23-CE12-0014-02).

## Author contributions

**Vinciane Piveteau**: Data curation; Investigation; Writing—review and editing. **Chloé Dupont**: Data curation; Investigation. **Hossein Salari**: Formal analysis; Investigation; Visualization. **Agnès Dumont**: Supervision; Investigation. **Jérôme Savocco**: Investigation. **Daniel Jost**: Formal analysis; Supervision; Funding acquisition. **Aurèle Piazza**: Conceptualization; Data curation; Formal analysis; Supervision; Funding acquisition; Investigation; Visualization; Writing—original draft; Project administration; Writing—review and editing.

Source data underlying figure panels in this paper may have individual authorship assigned. Where available, figure panel/source data authorship is listed in the following database record: biostudies:S-SCDT-10_1038-S44318-026-00748-6.

## Disclosure and competing interests statement

The authors declare no competing interests.

# Expanded View Figures

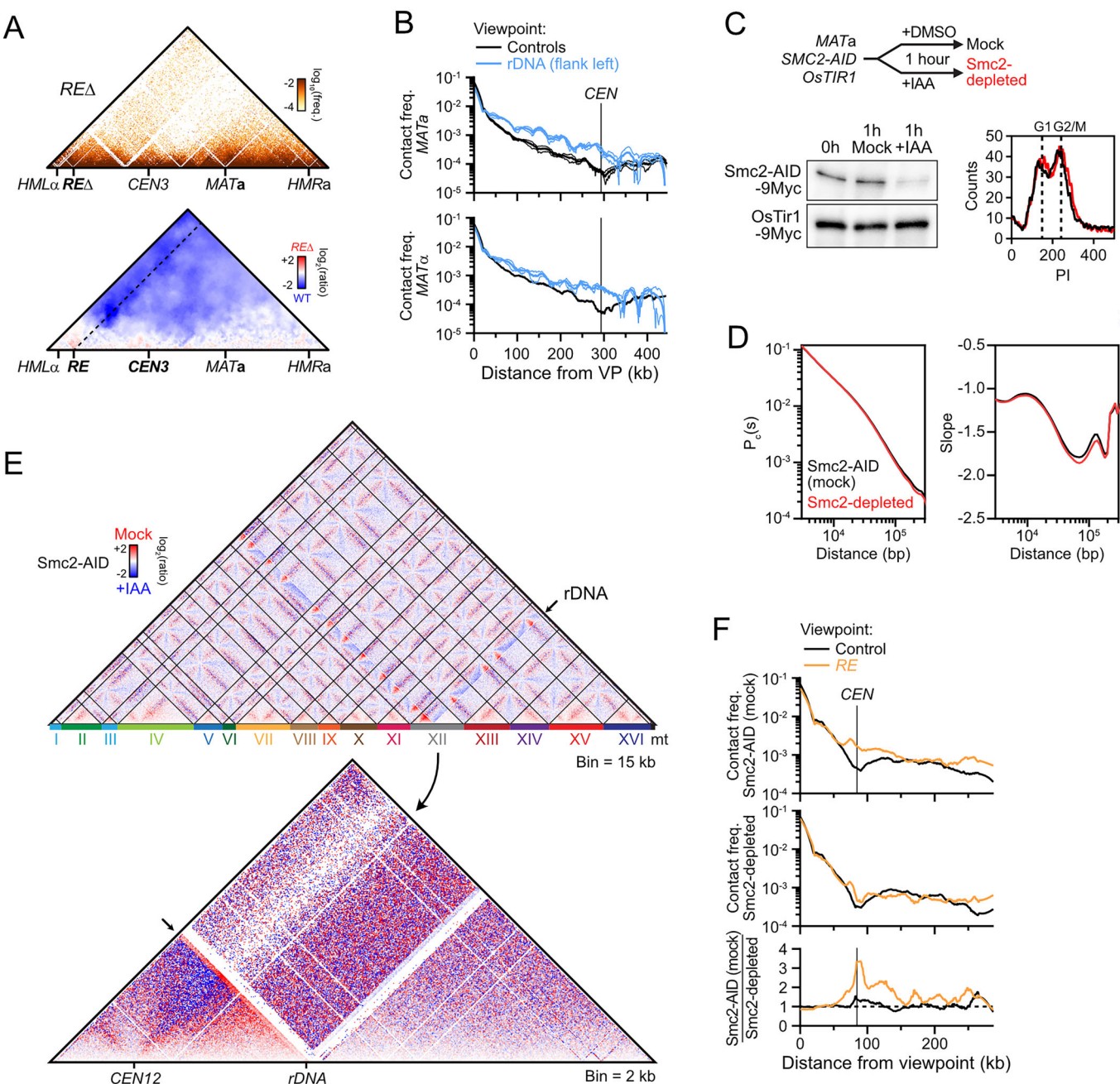

**Figure EV1. Condensin- and *RE*-dependent contact stripes in *MATa* cells.**

(related to Fig. 1). (**A**) Top: Hi-C contact map of chr. III in a *RE*-deleted *MATa* strain (APY1548). Bin: 1 kb. Bottom: Ratio map over a WT strain. Data show $n = 1$ biological replicate. (**B**) 4C-like contact profile of the left *rDNA*-flanking region (blue) and of the average of 6 control sites (black) as viewpoints in *MATa* and *MATα* cells, from Hi-C data in Fig. 1A. (**C**) Smc2 depletion scheme, Western blot validation, and FACS profiles of Smc2-AID (mock) and Smc2-depleted cells. (**D**) Probability of contact as a function of the genomic distance (Pc(s)) and its derivative in Smc2-AID (mock) and Smc2-depleted cells. (**E**) Ratio maps of the whole genome (top) and chr. XII (bottom) in cells proficient and deficient for condensin. (**F**) Top: 4C-like contact profiles of the *RE* and of the average of 6 control sites in Smc2-AID-tagged (mock) and Smc2-depleted samples, from Hi-C data in Fig. 1C. Bottom: Ratio of *RE* and control 4C-like profiles of Smc2-AID (mock) over Smc2-depleted samples. Data show $n = 1$ biological replicate.

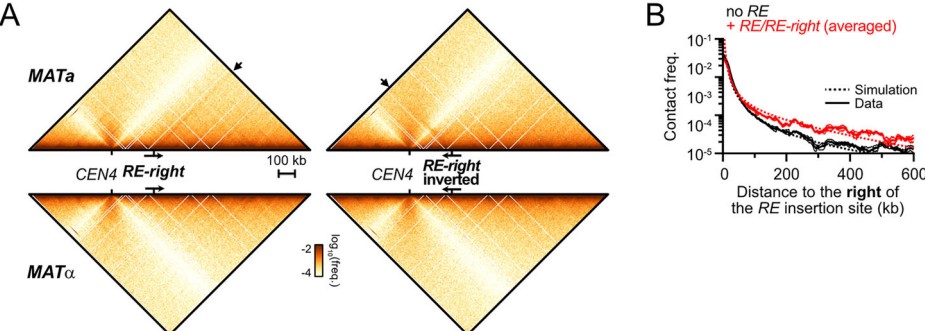

**Figure EV2. Condensin loop extrusion properties.**

(related to Fig. 2). (A) Top: Hi-C contact maps of chr. IV in *MAT*a and *MAT*α cells bearing the *RE-right* construct at position 680 kb either in the forward (APY1850 and APY1852) or inverted (APY2058 and APY2060) orientation. Hi-C maps are binned at 5 kb. Data show $n = 1$ biological replicate. (B) Observed and simulated 4C-like profiles using chr. IV 680 kb as a viewpoint, either unmodified ("no *RE*" black), or upon insertion of the *RE* or the *RE-right* constructs (data averaged). From data in Fig. 2C. The ratio of the "+*RE*" profiles over the "no *RE*" profiles gives rise to the normalized data presented in Fig. 2G.

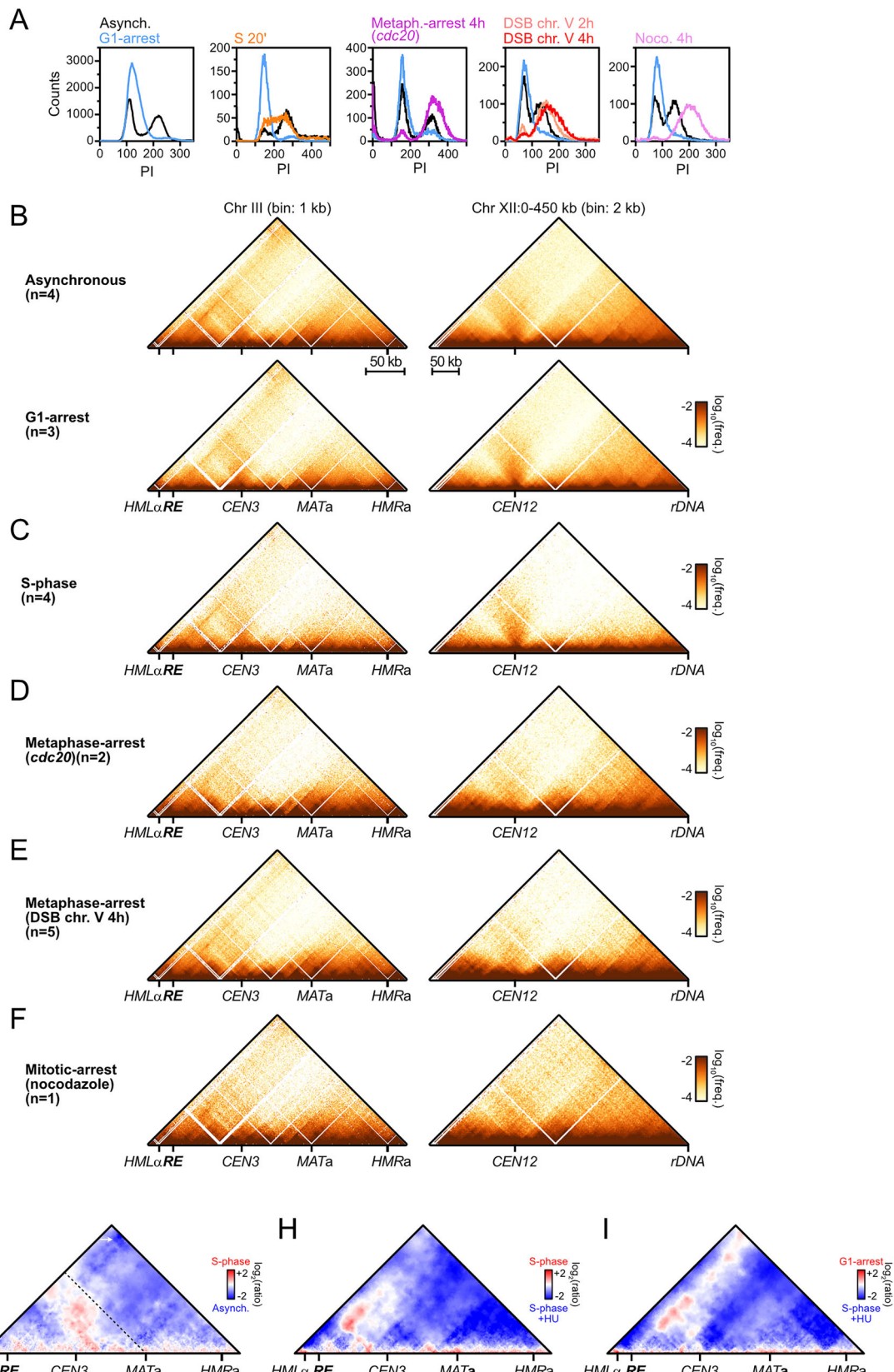

◀ **Figure EV3.  Regulation of loop extrusion by condensin across the cell cycle.**

(related to Fig. 3). (**A**) FACS profiles of the different cell-cycle stages studied here. (**B–F**) Hi-C maps of chr. III and chr. XII:0-450 kb (**A**) upon G1-arrest (APY266), (**B**) during S-phase (APY539 and APY607, merged), (**C**) upon metaphase-arrest due to *CDC20* repression (APY537), (**D**) upon DDC-induced metaphase-arrest due to formation of a single unrepairable HO-induced DSB on chr. V (APY266), and (**E**) upon mitotic-arrest in the presence of nocodazole (APY266). All cells are *MAT*a. The number of biological replicates (n) is indicated in each panel. Bin: 1 kb (chr. III) or 2 kb (chr. XII). (**G**) Ratio map highlighting the changes to chr. III structure in S-phase vs. asynchronous *MAT*a cells. From data in (**B**, **C**). (**H**) Same as (**G**) in untreated vs. HU-treated cells in S-phase. From data in (Jeppsson et al, 2022). (**I**) Same as (**G**) in G1-arrested vs HU-treated S-phase cells. From data in (Jeppsson et al, 2022).

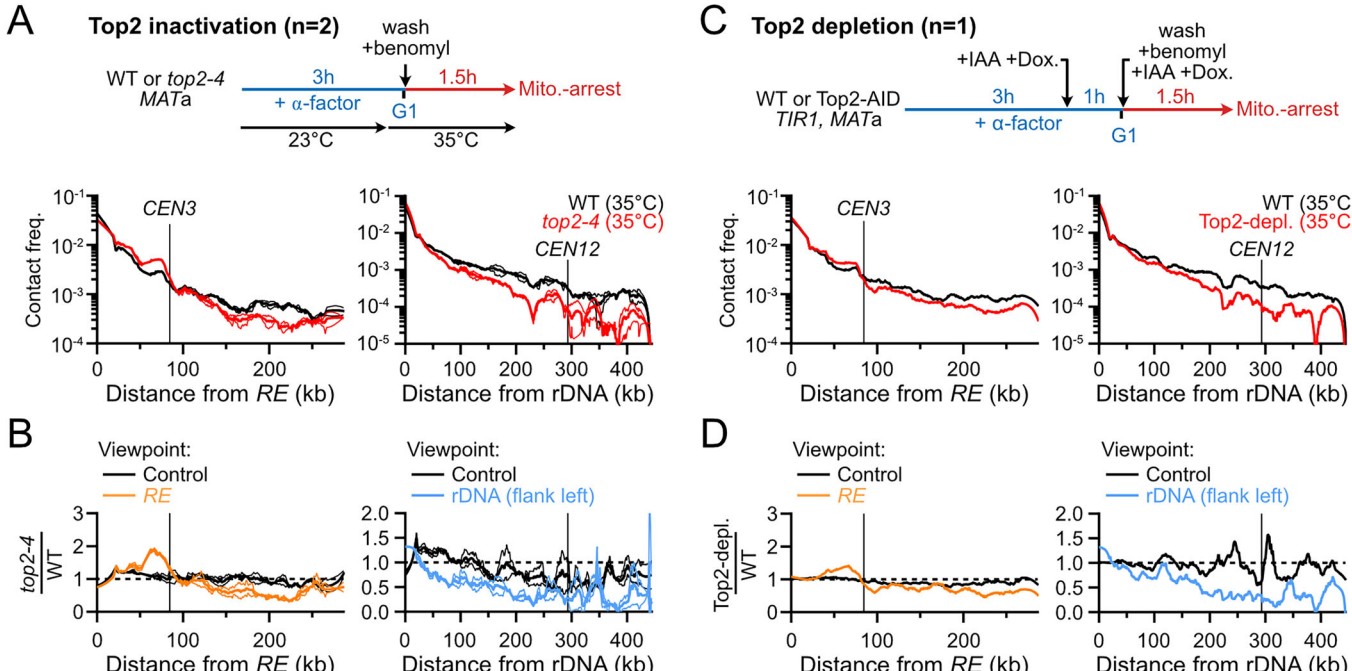

**Figure EV4. Loop extrusion by condensin is compromised in Top2-deficient cells.**

(A) 4C-like profiles of the *RE* (left) and the rDNA (right) and their cognate control sites in a mitotic WT and *top2-4* strains after 1.5 h at restrictive temperature. Data show mean ± SEM of $n = 2$ biological replicates. (B) Ratio of 4C-like profiles for the *RE*, *rDNA* and their cognate controls sites in *top2-4* over WT cells. Data show mean ± SEM of $n = 2$ biological replicates. (C) 4C-like profiles of the *RE* (left) and the rDNA (right) and their cognate control sites in mitotic WT and Top2-depleted strains. Data show $n = 1$ biological replicate. (D) Ratio of 4C-like profiles for the *RE*, *rDNA* and their cognate controls sites in WT and Top2-depleted cells. Data show $n = 1$ biological replicate. (A–D) All data are from (Jeppsson et al, 2024).

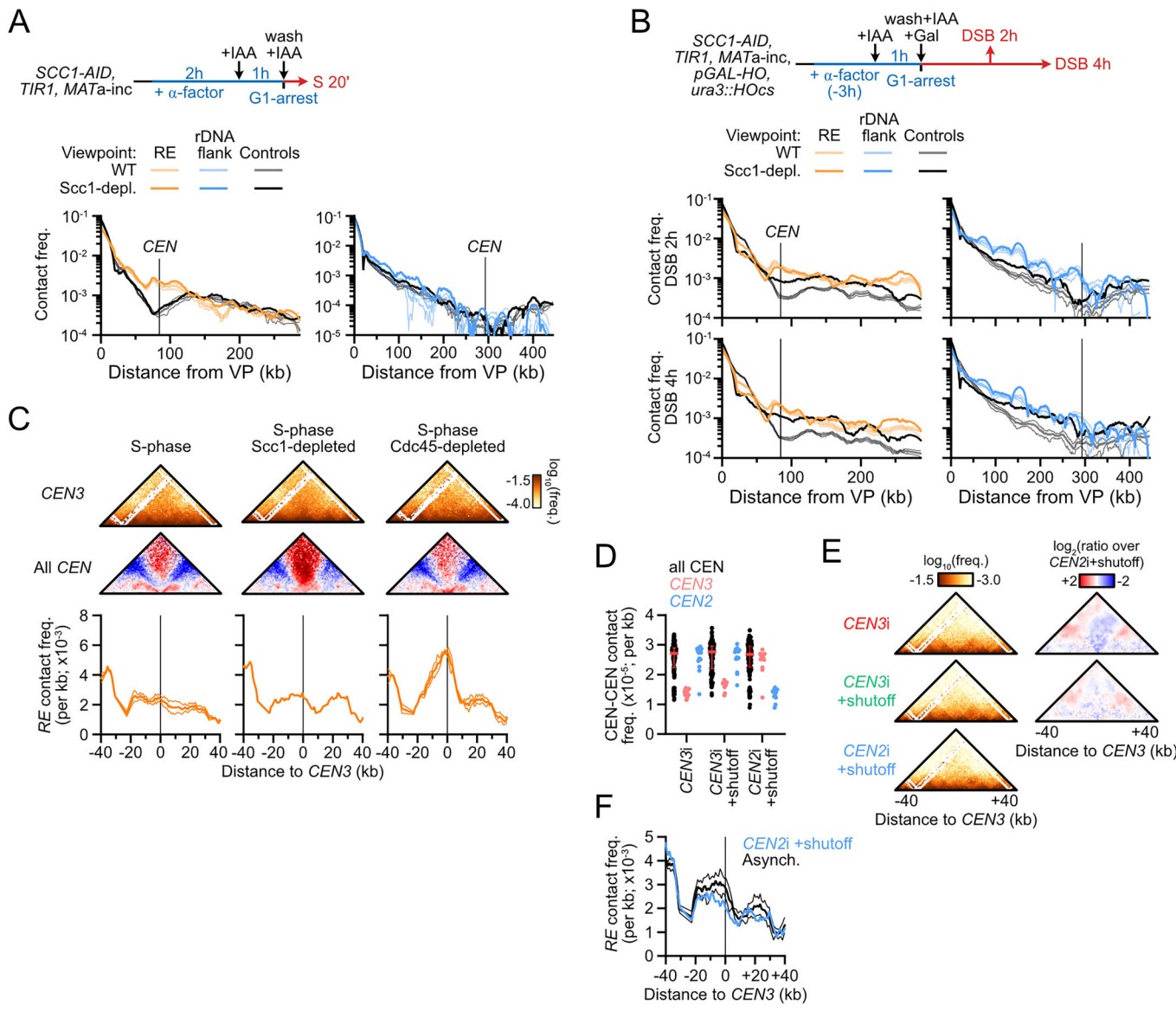

**Figure EV5. The centromere stalls condensin translocation in a kinetochore-dependent manner.**

(related to Fig. 4). (A) Loss of Scc1 does not rescue condensin-mediated loop extrusion in S-phase. Top: Scheme for Scc1-AID depletion prior to S-phase release. Bottom: 4C-like profiles at the *RE* and the left *rDNA*-flanking region and their corresponding control sites in WT and Scc1-depleted cells. *n* = 2 and 1 biological replicates, respectively. Data are from (D'Asaro et al, 2025). (B) Loss of Scc1 does not affect condensin-mediated loop extrusion in metaphase cells. Same as (A), but with Hi-C performed in cells arrested in metaphase upon formation of an unrepairable DSB on chr. V. Note the elevated baseline, particularly at 4 h post-DSB induction. *n* = 4 and 1 biological replicates for WT and Scc1-depleted cells, respectively. Data are from (Dumont et al, 2024). (C) Condensin roadblock at *CEN3* in S-phase in WT, Scc1-depleted, and Cdc45-depleted cells. Top: Hi-C maps of the *CEN3*-surrounding region. Middle: Aggregated contact maps of all centromeres. Bottom: *RE*-contact stripes. *n* = 2, 1, and 1 biological replicates for WT, Scc1-depleted, and Cdc45-depleted cells, respectively. Data are from (D'Asaro et al, 2025). (D) Inter-chromosomal contact frequency between all centromeres (black), between *CEN3* and other centromeres (pink), and between *CEN2* and other centromeres (blue) following transcription-mediated *CEN3* or *CEN2* inactivation. Each point represents a CEN-CEN contact frequency. Bars show median ± inter-quartile range. Data are from Fig. 4E, with each condition corresponding to *n* = 1 biological replicate. (E) Left: Hi-C contact maps of the *CEN3*-surrounding region (bin: 1kb). Right: ratio maps of the *CEN3*-surrounding region in *CEN3*-inactivated cells over control *CEN2*-inactivated cells. (F) 4C-like profiles with the *RE* as a viewpoint in WT and *CEN2*-inactivated strains. Data show mean ± SEM of *n* = 4 and 1 biological replicates, respectively.

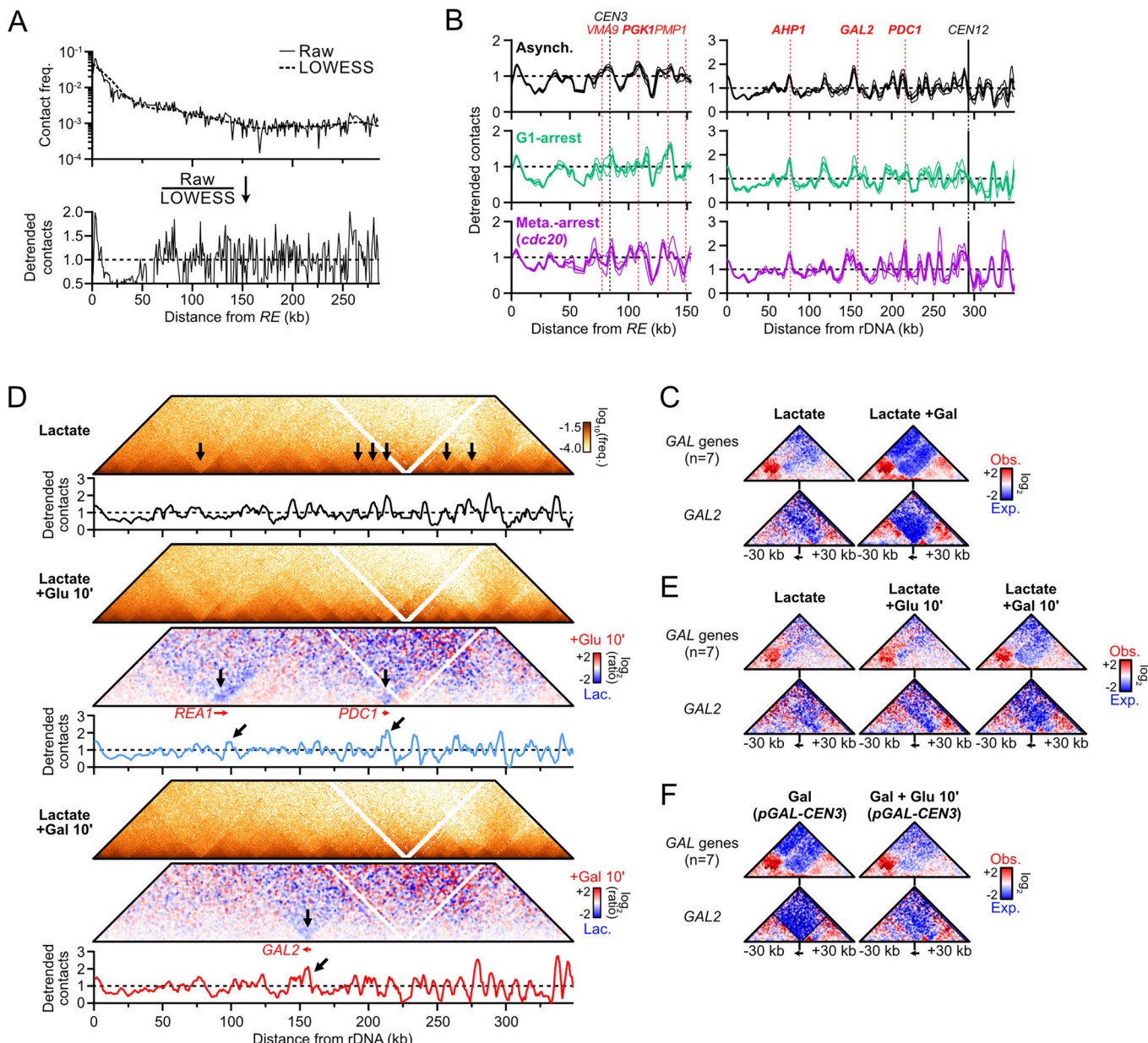

**Figure EV6. Highly transcribed RNA PolII-dependent genes stall condensin translocation.**

(related to Fig. 5). (**A**) Rationale for raw contact detrending over the LOWESS regression. (**B**) Detrended *RE* and *rDNA*-flanking contacts in asynchronous, G1-arrested, and metaphase-arrested cells grown in the presence of galactose and in the absence of glucose. Data are the same as in Figs. 1A and 3A,C. Highly transcribed genes present at the major peaks are indicated. (**C**) Observed over expected ratio maps aggregated at all *GAL* genes (top) and at *GAL2* (bottom) in galactose- and glucose-containing media. (**D**) Correspondence between high transcription (visible as discrete borders in the Hi-C map) and condensin loop extrusion pausing in lactate media and upon glucose or galactose addition for 10 min. *n* = 1 biological replicate each. (**E**) As in (**C**), from data in (**D**). (**F**) As in (**C**), upon glucose addition in galactose-containing media. From data in Figs. 4E and 5D.

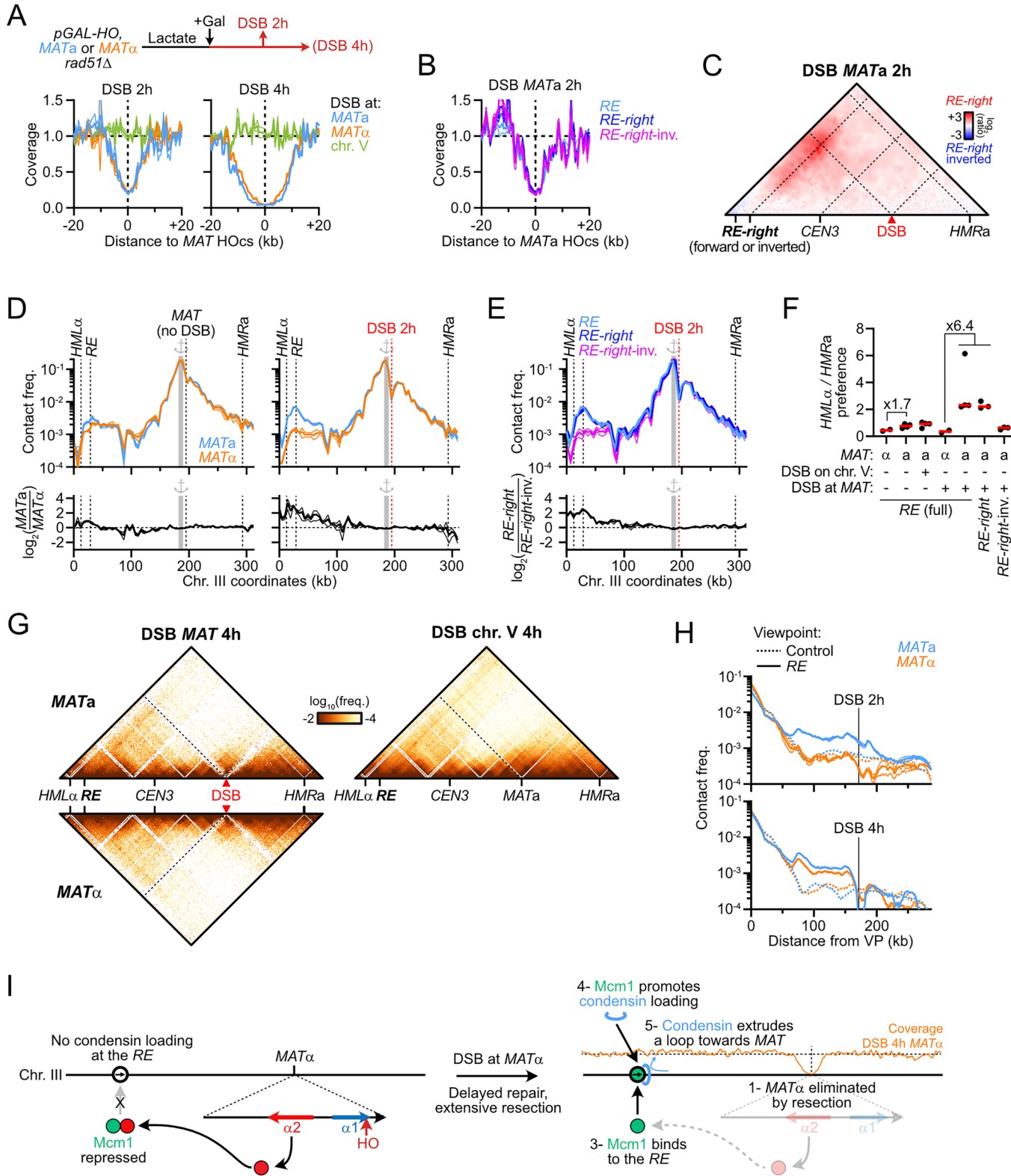

◄ **Figure EV7.  DSB formation at *MAT*a blocks condensin translocation and creates a *RE*-DSB loop.**

(related to Fig. 6). (**A**) Coverage from Hi-C reads at 2 and 4 h post-DSB induction at *MAT* in *MAT*a and *MAT*α cells. Control cells with a DSB on chr. V show no loss of coverage at *MAT*. From data in Fig. 6A. (**B**) Coverage from Hi-C reads at 2 h post-DSB induction at *MAT*a in cells with *RE* variants. From data in Fig. 6D. (**C**) Ratio map of cells with the *RE-right* over the *RE-right*-inverted construct 2 h post-DSB induction at *MAT*a. From data in Fig. 6D. (**D**) Top: 4C-like profiles with the 10 kb region left of *MAT* as a viewpoint in asynchronous cells and in cells 2 h post-DSB induction. Bottom: Log₂ ratio of profiles in *MAT*a over *MAT*α cells showing specific enrichment of contact between *MAT*a and the *HMLα-RE* interval. From data in Figs. 1A and 6A. (**E**) Top: 4C-like profiles with the 10 kb region left of *MAT* as a viewpoint in *MAT*a cells bearing different *RE* variants 2 h post-DSB induction. Bottom: Log₂ ratio of 4C profiles in *RE-right* over *RE-right*-inverted-containing cells. From data in Fig. 6D. (**F**) Quantification of the preference for *MAT* interaction with *HML* vs. *HMR*. Black data points show individual biological replicates (*n*), and the red bar shows the median. (**G**) Hi-C contact maps in *MAT*a and *MAT*α *rad51Δ* cells (APY1267 and APY1264, respectively) at 4 h post-DSB induction at *MAT*. Data show *n* = 1 biological replicate each. A *MAT*a cells with an unrepairable DSB on chr. V (APY266) is shown for comparison 2 h post-DSB induction (*n* = 5 biological replicates). Bin: 1 kb. (**H**) 4C-like profiles with the *RE* (or 6 control sites) as a viewpoint, from data in (**G**) and Fig. 6A. Data show mean ± SEM. (**I**) Model for the reactivation of condensin-mediated loop extrusion from the *RE* and establishment of a *RE*-DSB loop upon defective repair of a DSB at *MAT*α.

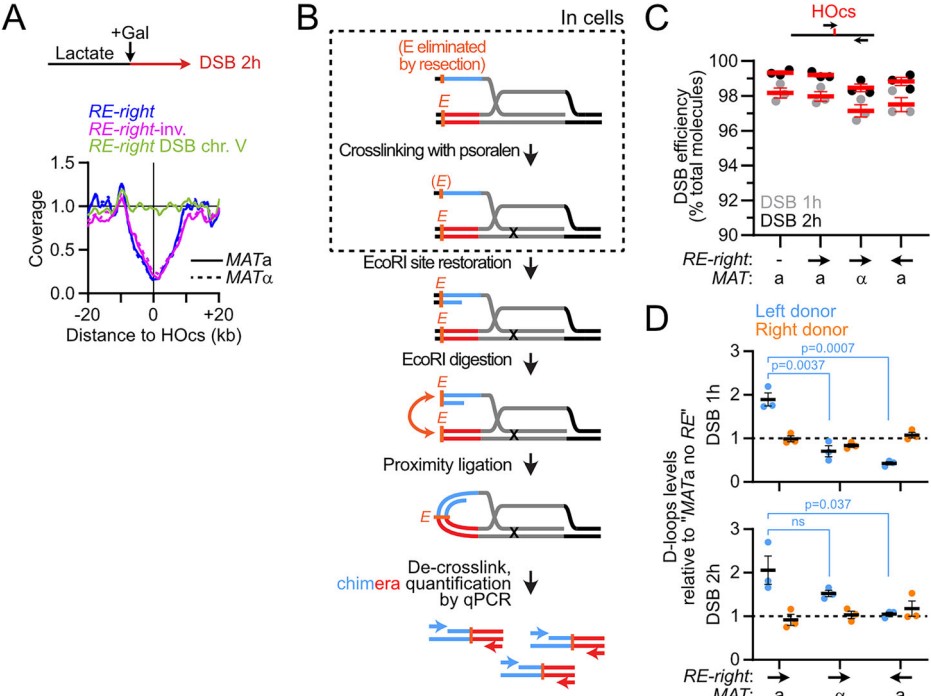

**Figure EV8.   The *RE*-DSB loop is portable and promotes *RE*-proximal homology search.**

(related to Fig. 7). (**A**) Coverage from Hi-C reads at 2 h post-DSB induction at the HOcs on chr. IV in *MAT*a and *MAT*α cells bearing the *RE-right* in forward or inverted orientation. Control *MAT*a cells with a DSB on chr. V show no loss of coverage at that site. From data in Fig. 7A. (**B**) Rationale of the D-loop Capture assay. (**C**) Quantification of DSB formation at 1 and 2 h post-induction. Data points show individual biological replicates (*n*). Mean ± SEM are shown in red. (**D**) D-loops levels expressed relatively to that measured in the *MAT*a strain without *RE* assayed in parallel. From data in Fig. 7D. Data show individual biological replicates (*n*) as well as mean ± SEM. *P* values were obtained using a Student *t* test. None of the comparisons for the right donor are significant.

