## [Peer Review File · The EMBO Journal]

Condensin loop extrusion properties, roadblocks, and role in recombination homology search in *S. cerevisiae*

Vinciane Piveteau, Chloé Dupont, Hossein Salari, Agnes Dumont, Jérôme Savocco, Daniel Jost, and Aurèle Piazza

Corresponding author(s): Aurèle Piazza (aurele.piazza@ens-lyon.fr)

Review Timeline:

Pre-submission Date:	8th Aug 25
Editorial Decision:	8th Aug 25
Revision Received:	19th Dec 25
Editorial Decision:	12th Feb 26
Revision Received:	16th Feb 26
Accepted:	27th Feb 26

Editor: Hartmut Vodermaier

Transaction Report: The first round of review of this manuscript was performed at another journal.

Dear Aurèle,

Thank you again for your interest in transferring a revised version of your manuscript on condensin-mediated loop extrusion to The EMBO Journal. As discussed, we would be happy to consider it for publication, pending positive re-review of specific aspects by some of the original referees and/or additional arbitrators of our own choice. Furthermore, we agreed that it would be preferable to delay resubmission until the ongoing functional assays of MAT switching efficiency would be finalized - with the study of course being covered by our scooping protection policy during that time. Below, you will find a link to upload your revised manuscript as soon as it is ready; at that stage, it would be great if you already adapted it as much as possible to EMBO Journal formatting guidelines, as this would strongly facilitate the editorial process at the time of resubmission:

<https://www.embopress.org/page/journal/14602075/authorguide#researcharticleguide>

<https://www.embopress.org/page/journal/14602075/authorguide#submissionofrevisions>

Please do not hesitate to contact me with any updates regarding the process of your work, the envisioned resubmission date, or the status of the competing work.

With kind regards,
Hartmut

To submit your revised manuscript, please follow this link.

Link Not Available

Reviewer #1 (Remarks to the Author):

The researchers address how the mechanism by which 3D genome is organized influences recombination, using the recombination enhancer in yeast as their system. First, the researchers used ectopic insertion and deletion of RE and condensin depletion experiments to establish that condensin is responsible for the directional interactions emanating from RE. These are well controlled experiments and interpretations are clear. Next, the researchers use modeling to conclude that condensin is loaded at RE and processivity is reduced during S phase. Next the authors study various blocks of condensin using published and new data. Finally, the authors show that in a *rad51* mutant, DSB at *Mata* is a block to one sided loop extrusion that started at RE. Experiments demonstrate that RE is a loading site for directional loop extrusion and bring *MAT* and RE together. There are 2 issues with the manuscript as it stands. 1) conclusions on SMC5/6 are not well supported due to lack of quantification and consideration of alternative explanations. 2) the current model does not considerably increase our current understanding in how RE functions as it does not test how proximity relate to donor preference and does not consider alternative hypotheses for RE function.

We thank this reviewer for his/her careful assessment of our work. We have conducted several new experiments and analyses suggested by this and other reviewers to strengthen the functional claims that condensin-mediated *RE*-DSB loop formation contributes to regulate donor selection during *MAT* switching.

Major comments:

1) The authors state that “The loop extruding *Smc5/6* complex (Pradhan et al, 2023) mainly associates with chromatin following S phase (Jeppsson et al, 2014).” I caution the authors from making the claim of loop extrusion for *Smc5/6*, because the in vitro loop extrusion activity was demonstrated in vitro but unlike cohesin and condensin, where HiC experiments in various systems are consistent with the loop extrusion hypothesis, similar experiments have not been performed for *Smc5/6*. The Hi-C analysis in SMC5/6 depletion focused on loop intensity, and can not conclude on processivity, which is the basis of the loop extrusion hypothesis i.e. progressive enlargement of loops. For instance, progressive loop enlargement was tested for cohesin using conditions that alter its processivity (e.g. *wapl*) and quantifying average loop sizes in derivative of distance decay curves. For condensin, in conditions where there is a loading site, e.g. *rex* sites in *c elegans*, orthologous methods to Hi-C such as ChIP-seq corroborated the progressive movement of the complex. No such experiments has been done for SMC5/6, to the best of my knowledge, thus it is early to make a statement such as “loop extruding SMC5/6 complex”.

We agree with this comment that the work by (Pradhan et al, 2023) does not provide direct evidence for loop extrusion by *Smc5/6* in cells.

2) The authors state that “However, *Smc5/6* depletion partly relieved the barrier for loop extrusion observed in the absence of *Top2* (Fig. 4C and S4D).” This statement is not well supported by the data. The changes are very small. To make this conclusion the authors need to quantify the differences between the single and double depletions and demonstrate that the quantitative difference is reproducible across biological replicates. In addition, while the authors

suggest the SMC 5/6 is the block, there is an alternative possibility that depleting SMC 5/6 affect the number of physiology of the replication forks themselves. I think the author’s interpretation require additional controls to eliminate alternative interpretations.

This is similar to comment #6 by reviewer 3.

Data presented in Fig. 4 were re-analysis of previously published datasets that were controlled for in Jeppsson et al. Mol. Cell 2024. In this work, two independent strategies were used to deplete/inactivate both Smc5/6 and Top2, each time with matching controls:

- *top2-4* heat inactivation was performed in duplicate.
- *top2-AID* depletion was performed once.
- *smc5-AID smc6-AID* co-depletion was performed in duplicate.
- *smc6-AID nse4-AID nse5-AID* co-depletion was performed once.

The results from these different conditions and replicates are highly concordant and can be used to establish the following facts: Smc5/6 is not a roadblock for condensin; and that Top2 promotes loop extrusion by condensin.

The combination of Smc5/6 and Top2 depletion was performed only once and we agree with this reviewer that a lack of biological replicate for this double-depletion experiment (although accepted for publication elsewhere) is insufficient to draw firm conclusions.

Consequently, and since the Smc5/6 datasets show no effect with respect to condensin loop extrusion in and on itself, we decided to shrink this re-analysis of the Jeppsson et al. data, by presenting only the effect of a Top2 deficiency in an updated result section “Condensin-dependent loops are shortened in the absence of Top2” and in an updated **Fig. EV4**.

3) Unidirectional loop extrusion from RE to the MAT juxtapose the two sites but does not yet explain how it increases recombination with HML, other than reducing distance. The authors

should consider discussing the implications of their model on the recombination bias with HML and HMR. For instance, they can computationally model if juxtaposing RE and MAT is enough to achieve the % bias occur based on % contacts between MAT and the left of the RE in Fig 7E? In addition, the authors should discuss how their model works with an alternative activity proposed for RE: that the recombination proteins “enter” at RE because activated transcription opens DNA this site and such that homologous recombination proteins can search for homology towards HML: PMIDs 16166630 and 16135790 If “entry” model is true, recombination proteins march the distance between the RE and HML alpha. Without linking increased contacts to the function of RE, although shown with beautiful ectopic insertions and deletions, the current model in Figure 7H does not push our current understanding of RE increasing contact frequency MAT with the left arm of the chromosome based on Hi-C maps (eg PMID 37058545).

Functional end-point assays established a partial dependency of α -to- α switching on condensin in two previously published papers we referenced extensively (Li .. Smith PLoS Genetics 2019 and Dinda.. Smith PLoS Genetics 2023). Specifically, they addressed the role of the exact condensin loading site with a narrow 100 bp deletion in the “right” part of the RE.

The studies this reviewer refers to relates to the “left” part of the RE (also refer to as the “700 bp RE”) whose function in MAT switching depends on Fkh1. Indeed, Fkh1 (and more specifically its FHA domain) is a major contributor to donor selection in MAT α -to- α switching (Wu & Haber Cell 1996, Coic et al. MCB 2006, Li .. Haber PLoS Genet 2012) and even works in *trans* by interacting with the Rad51-ssDNA filament (see Lee .. Haber PNAS 2016, Roy .. Steinmetz Nat Biotech 2018 and our own work Dumont et al. Mol Cell 2024).

To address this reviewer concern we conducted several additional experiments and analyses that support our model.

First, we showed that the RE-DSB loop formed on chr. III occurs independently of the Fkh1-containing part of the RE (*RE-right* construct) and requires the loop extrusion to proceed towards MAT (abolished upon inversion of *RE-right*), see new **Fig. 6D-E**.

Second, following the suggestion by this reviewer, we performed additional analysis that quantify the contacts between *MAT* and *HML/HMR* before and after DSB formation (new **Fig. EV7D-F**). Strikingly, formation of a DSB led to a 6-fold preference for *MAT* contacting *HML* over *HMR* in *MAT* α cells compared to *MAT* α cells. By contrast, this difference is only 1.7-fold in the absence of DSB. Again, this preference is independent of the Fkh1-containing left part of the *RE*, and depends on the orientation of loop extrusion towards the DSB. Consequently, the juxtaposition of *MAT* α and *HML* α is primarily established upon DSB formation. This observation supports our model that the relevant structure in promoting *MAT* α -to- α switching is the *RE*-DSB loop formed upon arresting condensin at the DSB rather than the heterogenous folding of chr. III previously described in asynchronous cells (eg Belton .. Dekker Cell Rep 2015).

Third, we introduced the part of the *RE* lacking the Fkh1 domain (*RE-right*) on chr.IV and induced a DSB 165 kb downstream. This ectopic system recapitulated the *RE*-DSB loop observed on chr. III in an otherwise WT background, only in *MAT* α cells (new **Fig. 7A-B**). Again, this loop required the *RE* to be oriented so that loop extrusion proceeds towards the DSB, as inverting the *RE-right* construct results in a contact stripe running in the opposite orientation (see new **Fig. 2B** and **EV2A**) and abolish the *RE*-DSB loop formation. Consequently, the right, condensin-containing part of the *RE* is necessary and sufficient for *RE*-DSB loop formation independently from the “left” Fkh1-dependent module of the *RE* that acts in *trans*.

Finally, we determined the effect of this *RE*-DSB loop in biasing homology search towards a *RE*-proximal donor (such as *HML α*). To this end, we introduced two competing donors in our ectopic system on chr. IV so as to emulate the distances between *HML*, *MAT* and *HMR*, and quantified D-loops at these two donors by D-loop Capture (Piazza et al Mol. Cell 2019) (new **Fig. 7D-E**). This readout enabled us to ascertain an effect of this condensin-dependent loop at the homology search step, and not at subsequent steps of the HR pathway, unlike the previously used end-point assays scoring repair product formation. Presence of the *RE-right* construct biased donor identification by promoting D-loop formation at the *RE*-proximal donor. This effect was *MAT α* specific and depended on the orientation of loop extrusion towards the DSB. Consequently, the condensin-mediated *RE*-DSB loop promotes homology search and identification near the loop anchor, *ie* the condensin loading site.

These results demonstrate that (i) a stereotypical *RE*-*MAT* loop forms when a *RE*-bound loop-extruding condensin runs into a DSB at *MAT*, independently of the Fkh1-binding module of the

RE, (ii) that this loop juxtaposes *MATa* with its *HMLα* donor several fold more than in the absence of a DSB, (iii) that this two-components *RE*-DSB looping system is portable, as it can be reconstituted ectopically on chr. IV, and (iv) functionally, that this loop stimulates homology identification of the *RE*-proximal donor, thus biasing donor preference in a competitive donor context.

These observations do not support the suggestion by this reviewer of a transcription-mediated chromatin opening from which HR protein would slide towards *HMLα*. Indeed, this function has been ascribed to the recruitment of Fkh1 to the left part of the *RE* (Ercan et al. MCB 2005), but we show here that the phenotypes observed occur independently of the Fkh1-containing part of the *RE*, and that the DSB-induced *MATa*-*HMLα* juxtaposition occurs independently of Rad51 (Fig. 6A-E). Furthermore, introduction of the *RE*-right part lacking the Fkh1 binding module near a donor on chr. IV was sufficient to stimulate its identification, in a manner that depended on its orientation towards the DSB. Our experiments thus circumscribe a Fkh1-independent, condensin-dependent role of the *RE* in biasing donor selection during HR repair. We discuss the interplay between these two modes of actions of the *RE* in an expanded paragraph in the discussion L436-448:

“MAT switching thus provides a model to study the basic mechanism that establishes selective interactions between chromosomal segments. Biasing for HMLα usage in MATa cells depends on two main modules in the RE: a condensin-loading module whose deletion reduces donor bias by 2-fold (Li et al, 2019); and a Fkh1-binding module active only in G2/M whose deletion almost abolishes HMLα usage (Wu & Haber, 1996; Sun et al, 2002; Coïc et al, 2006b) (Fig. 2A). This major role of Fkh1 at the RE specifically depends on its FHA domain (Li et al, 2012) and can act in trans (Coïc et al, 2006b; Lee et al, 2016) by recruiting the Rad51-ssDNA filament (Dumont et al, 2024; Renkawitz et al, 2013). We thus propose that the more modest role of condensin in promoting MATa-to-α switching may be to promote the establishment of the first link between the RE and the broken MATa locus, whose maintenance would subsequently be handed over to Fkh1 and the Rad51-ssDNA filament. This two-step scenario for establishing specific long-range interactions in cis along chromosomes thus consists of an initial phase of moderate specificity modulated by intrinsic properties of the SMC complex and its roadblock followed by a maintenance phase dependent on the protein-protein and protein-DNA affinities of two (or more) cognate DNA-bound factors.”

Minor comments

1- Line 168 typo “stongly”

Thanks for noticing.

2- Line 251 “nor the increased rDNA-GAL2 looping (Fig. 6B-C and S7C)” did the authors mean S6C?

Thanks for noticing, we meant S5C (now EV6C).

3- In figure 5, can the authors mark where the “blocks” are? It is difficult to read from the small Hi-C maps. Perhaps a cartoon of the chromosome folding can also help. Enlarging as much as possible would help as well. It is also unclear to this reviewer, why the authors conclude these are blocks. Aren't condensins loaded at centromeres?

The block is at the “0” coordinate, as plots are centered on *CEN3*. The block is visible on the 4C-like profile as a peak of contacts with the viewpoint (*i.e.* the *RE*). The Hi-C maps are shown to enable the reader to compare the status of the block on the 4C-like profile to the conformation of the pericentromere at that site, which changes drastically between G1 and metaphase and in conditions that undermine the exertion of spindle tension (**Fig. 4A-C, F**).

The interpretation that condensin is loaded at the centromere has been inferred from ChIP data (eg D’Ambrosio et al Genes Dev 2008, Leonard et al. Cell Rep 2015), that formally only reveal sites of condensin enrichment. Since condensin can travel long distances, as we show here, such enrichments can also represent stalling sites.

4- The authors state in line 283 “This re-establishment occurred after the elimination of the *MAT α* genes by resection (Fig. 7A).”. I do not see any data or references showing that the resection did not happen in the conditions used?

We apologize for the confusion. We meant that the establishment of a *RE-MAT α* loop following DSB formation only occurred after resection had eliminated the *MAT α* -encoded genes. The resection profile that shows full elimination of the *MAT* region 4h post-DSB induction was shown in Fig. S6A (and now zoomed into in **Fig. EV7A**).

We modified the sentence as follow L310 “*This loop only formed after the elimination of the *MAT α* genes by resection*”. The rationale for loop re-establishment, supported by data from (Dinda et al. PLoS Genet 2023) is depicted in **Fig. EV7I**.

Reviewer #2 (Remarks to the Author):

In this study, researchers investigate the formation and regulation of condensin-dependent chromatin loops in *Saccharomyces cerevisiae* (budding yeast), focusing on two specific chromosomal contact stripes, one at the Recombination Enhancer on chromosome 3 (only in MATa cells) and the other at the left side of the ribosomal array on chromosome 12. Using high-resolution Hi-C, they demonstrate the dependency of these stripes on condensin. Through relocating the RE to chromosome IV, they also demonstrate that the RE is both necessary and sufficient for establishing a condensin-mediated stripe.

Moreover, the authors modelled condensin dynamics to estimate condensin's processivity and density along chromatin. They also investigated extrusion on cell-cycle stages and show that extrusion is impeded by replication forks in S-phase. In addition they studied other potential roadblocks for extrusion demonstrating that genomic features like centromeres and highly transcribed RNA PolII genes, can indeed act as obstacles to processive extrusion. Notably centromere barriers required kinetochore components, and are shown to be independent of spindle tension or cohesin, while transcription-induced extrusion roadblocks are shown to be very dynamic, and rapidly reversed correlating with transcriptional activation and repression.

The authors also investigated extrusion in the absence of topoisomerase II (Top2) and showed defects in this condition suggesting that Top2 promotes condensin extrusion, interestingly this could be reversed by inhibition of the Smc5/6 complex, suggesting that Smc5/6 engaged on DNA as a consequence of topological stress acts as an extrusion barrier.

Finally, the authors investigated the role of the condensin-dependent stripe on chromosome 3 in the process of mating type switching. They showed that a double-strand break (DSB) at MAT disrupts condensin-dependent extrusion and propose that this could bring the DSB into proximity with the HML α donor site hence facilitating the switch.

While this study is technically sound and provides interesting insights, it is largely descriptive, focusing on observations across various conditions without deeply engaging in mechanistic tests. The most compelling aspect is the suggested functional relevance of loop extrusion for MAT switching. However, in the final section, the authors mainly present observational data and speculate that loop extrusion might facilitate MAT switching, without directly testing this hypothesis.

To strengthen the study, I recommend that the authors conduct additional experiments to directly test the role of loop extrusion in MAT switching.

We thank the reviewer for his/her careful evaluation of our work. Indeed, our study provides (i) the first determination of the loop extrusion activity of condensin in cells (loading site, processivity, directionality, orientation) across the cell cycle, (ii) identify 4 different types of positional roadblocks of varying intensity and delineate their main requirements, and (iii) investigate the function of this loop extrusion activity in the process of mating-type switching, for which functional data already implicated condensin without providing a mechanistic basis (Li .. Smith PLoS Genetics 2019 and Dinda.. Smith PLoS Genetics 2023).

Given the density of the current manuscript, a deep mechanistic characterization of all the novel observations reported here appears beyond the scope of a single publication. However, in line with this and other reviewers' comments, we deepened the characterization of the loop extrusion properties of condensin and the mechanism by which it impinges on donor selection during recombinational DSB repair (see below).

1. The directionality of condensin extrusion starting at the recombination enhancer (RE) and the observation that inserting the RE on chromosome 4 generates a directional extrusion stripe suggest that this loading site might contain clues responsible for directionality. The authors should introduce the RE on chromosome 4 in reversed orientation and test whether this alters the direction of extrusion.

This is a very interesting point and we conducted the experiments suggested by the reviewer with two *RE* constructs: one containing the full *RE* and one containing only the condensin-binding right part.

Indeed, found that the inverting these constructs on chr. IV reversed the direction of the stripe (new **Fig. 2B** and **EV2A**). It shows that the directionality of loop extrusion is encoded in the *RE* sequence, independently from the Fkh1-binding domain of the *RE*.

Using the *RE-right* construct, we could also show that the directional loop extrusion towards the DSB is required for forming a *RE*-DSB loop, independently of the Fkh1 domain (new **Fig. 7A-B**).

2. Once confirmed that the orientation of extrusion is reversed by flipping RE on chr4, the hypothesis that loop extrusion facilitates MAT switching to MATα could be directly tested to demonstrate its physiological relevance, which is currently lacking. To test this, the authors would invert the RE site on chromosome 3, confirm extrusion in the opposite direction and measure the efficiency of MAT switching in this new strain by comparing it to the original strain with the RE in its standard orientation. The prediction is that the MAT DSB will repair less efficiently with HMLα (even a delay would be significant) and may instead promote repair with the nearby HMRα site. This would strongly support a physiological role for condensin-mediated loop extrusion in donor choice during MAT switching.

We thank the reviewer for this suggestion. Functional end-point assays already established a partial dependency of a-to-α switching on condensin in two previously published papers we referenced extensively (Li .. Smith PLoS Genetics 2019 and Dinda.. Smith PLoS Genetics 2023). Specifically, they addressed the role in MATα-to-α switching of the exact condensin loading site with a narrow 100 bp deletion in the “right” part of the RE.

To address the broader generality of this observation, we conducted a similar experiment to that proposed by this reviewer, but in our minimal ectopic system on chr. IV. Furthermore, we did not detect recombination product, but the D-loop intermediate, *ie* the first DNA joint molecule formed upon successful homology identification; a more direct readout of homology search than end product detection. To this end, we introduced two competing donors in our ectopic system on chr. IV so as to emulate the distances between HML, MAT and HMR, and quantified D-loops at these two donors by D-loop Capture (Piazza et al Mol. Cell 2019) (new Fig. 7D-E). Presence of the RE-right construct biased donor identification by promoting D-loop formation at the RE-proximal donor. This effect was MATα specific and depended on the orientation of loop

extrusion towards the DSB. Consequently, the condensin-mediated *RE*-DSB loop promotes homology search and identification near the loop anchor, *ie* the condensin loading site.

D

E

These results establish the role of condensin in homology search regulation in the specific case of *MAT* switching. We discuss the distinction between that mediated by another SMC, cohesin, in a final discussion section “*Cohesin and condensin loop extrusion properties differently regulate homology search*” (L449-472).

Reviewer #3 (Remarks to the Author):

Piveteau et al. investigated condensin-mediated long-range chromatin interactions and concluded that condensin acts as a loop extruder that can be blocked by multiple types of roadblocks. Their Hi-C data confirmed previous finding that condensin is required for Chr III folding involving the RE site used for mating type switch. Focusing a contact 'stripe' originating from RE in HiC data, polymer simulation was conducted assuming that condensin acts as a loop extruder in a scrunching mode, and best model fits were deduced for condensin loading vs. processivity. Subsequently, the effects of possible blockages of condensin loop extruding were examined, including replication forks, centromeres, highly transcribed genes, Top2, SMC5/6, and cohesin. This set of HiC analyses focused on both the RE strip signal in HiC data generated in this work or from previous studies and some rDNA HiC analyses. Finally, the effect of a DSB at the MAT locus on the RE-MAT looping was examined. The topic of the work is interesting and the RE strip can served a nice model to examine condensin mediated chromatin folding. However, we have several major concerns on data repeatability, interpretation, and conclusions. Writing is difficult to understand at places and figure presentations should be improved.

We thank this reviewer for his/her appreciation of our work. We added replicates and improved data presentation in line with his/her suggestions.

Major concerns:

1. Most conclusions of the manuscript were drawn based on single HiC dataset per experimental condition. This is particularly problematic for model fitting, deriving parameters such as loop extruding processivity and condensin loading density, and to compare two strains. Without

delineation of experimental error ranges, parameter deduction and model fit would not be convincing. Further, a polymer model based on a single type of data requires validation using other approaches. Given that condensin is a unidirectional loop extruder, one could at least validate condensin density at its anchor sites by calibrated ChIP-seq, for example, examining changes before and after IAA treatment in Smc2-AID cells. This approach would help verify the condensin density predicted by the model.

This reviewer is incorrect in stating that most conclusions were drawn from single Hi-C datasets. Model parameters were determined on asynchronous cells (n=4) and along the cell cycle: G1-arrest (n=3), S-phase (n=2, now n=4), and multiple artificial and natural contexts for metaphase-arrest (*cdc20* n=2; DSB 2h n=4; DSB 4h n=5), where n is the number of biological replicates. Hi-C libraries and their correspondence to figure panels listed in **Table EV2** all originate from independent biological replicates. In instances in which a single replicate was performed, a similar condition consolidates the results and interpretation. See for example the introduction of the “full *RE*” or its right part (*RE-right*) on chr. IV (Fig. 1F and S1I, now **Fig. 2B** and **EV2A**), or the *chl4* mutant with and without nocodazole (now **Fig. 4F**), or *Sccl* depletion in metaphase and S-phase (Fig. 5D and S5D), nocodazole treatment with and without a DSB (Fig 5B; , nocodazole alone n=1, nocodazole + DSB 4h n=3), etc. We added replicates to DSB induction experiments at *MAT* (now n=2 to 4).

As requested by this reviewer, we plotted the inter-experimental variation (mean \pm SEM) in all figures.

The calibrated ChIP-seq will not allow to corroborate or infirm the density derived from the model, as it does not provide an absolute occupancy of condensin per DNA molecule in the cell population.

2. "Condensin-mediated looping on chromosome III is a complex process involving at least half a dozen additional proteins that bind to different regions of the RE sequence. Whether these factors contribute to HiC pattern changes seen upon the loss of replisome assembly, centromeres, highly transcribed genes, Top2, SMC5/6 need to be addressed before reaching conclusions. One should discern if observed HiC contact changes in these perturbation conditions reflect blockages (or lack of them) of condensin-mediate loop extrusion or rather reflect indirect consequences induced by changes of chromatin property.

We agree that genome-wide changes to chromatin property may confound the analysis of the contact stripe. That is why we specifically controlled for such changes by plotting contacts made at control loci alongside the stripe data in all figure panels. Control contact profiles were computed as described in the former Fig. S1B, now a main **Fig. 1B**, and as detailed in the Methods (L662-667): “For each viewpoint [i.e. *RE* or *rDNA*], 6 control viewpoints located at the same distance from a centromere and on chromosome arms of a similar length were selected, processed in the same way, and their contact frequency averaged. Control viewpoints for the *RE*, in the form *chr:coordinate(strand)*, were *V:66000(W)*, *VIII:21000(W)*, *XIV:713000(C)*, *II:153000(W)*, *XI:525631(C)*, *XIII:182646(W)*. Control viewpoints for the *rDNA*-flanking region were *IV:747000(C)*, *XIV:332000(W)*, *XIII:565000(C)*, *XV:624000(C)*, *XI:143000(W)*, *II:536000(C)*.” See also our point to comment #5.

The best-fitting extrusion parameters (density and processivity) are determined against the ratio of contacts over this baseline (see new **Fig. 2E-H**) as described in the Methods (L715-718):

“By adjusting the parameters of the model ($p, d, \{\alpha_i\}, \{l_i\}, \{\Delta l_i\}$), we aim to achieve the best fit with experimental observations. In summary, we compare the ratio of experimental contact frequency to its corresponding control with the theoretical prediction of $P(s)/P_0(s)$. By iteratively adjusting the model parameters, we identify the optimal values.” $P_0(s)$ is the contact frequency in the absence of loop extruders, described by: $P_0(s) = As^{-\gamma}$ (Halverson *et al*, 2014). Consequently, the model accounts for changes to chromatin properties unrelated to condensin-mediated loop extrusion.

At the exception of the metaphase-arrested cells depleted for Scc1 (**Fig. EV5B**), none of the conditions tested significantly affected the baseline.

3. Studies of chromatin loci contacts based on HiC data can be technically challenging for rDNA for several reasons. rDNA contains more than 100 copies of 9kb DNA sequences, and HiC data have limited ability to distinguish contacts amongst rDNA repeats and with surrounding regions. In addition, condensin is key to support 3D conformation of the entire rDNA array, and this role can contribute to the changes seen in rDNA contacts with the specific regions in HiC data, adding complexity in data analyses. Moreover, rDNA repeat numbers changes widely among strains, which making comparison between strains for rDNA contacts difficult. Further and perhaps different types of assays are needed to address these issues to discern the role of condensin in establishing rDNA contacts with other regions. Finally, the HiC data regarding condensin-mediated contact stripe at rDNA in this report was only shown as ratio map (Figure S1F) and the original contact maps should be provided. Nevertheless, this map appears to show that the 2 rDNA repeats, which were used to represent the entire rDNA array, make contacts with specific sites across each chromosome, and these interactions appear to be condensin-dependent. These interactions were not discussed in detail in the paper but should be included. The contact strip on ChrXII suggests that condensin loss reduces the contacts between rDNA and regions

located between rDNA and Tel12L, with the strongest effect seen before reaching CEN12. This pattern differs greatly from what is seen in the RE strip. This difference should be discussed.

As this reviewer points out, using the rDNA as a viewpoint is going to suffer from much analysis drawbacks. We circumvented this difficulty by analyzing the unique region immediately flanking the rDNA array, on the left side (*i.e.* on the *CEN12* side), which we denoted “rDNA (flank left)” in the annotation of all figures. The stripe is thus a proxy that reports on the behavior of the leftmost rDNA repeat. We now depict this subtlety in **Fig. 1D**.

We now show all the Hi-C maps in asynchronous cells and at different cell cycle phases in an updated **Fig. EV2B-F**. As pointed out above, the rDNA itself was excluded from the maps. In this view, the rDNA-flanking viewpoint corresponds to the rightmost bin. The reference genome was edited to contain only 1 rDNA repeat, as stated in the Methods section (L612-615): “*Hicstuff* pipeline was used to align pairs of reads [...] using a modified *S. cerevisiae* R64-2-1 reference genome in which the *HML*, *MAT*, *HMR*, *URA3*, *LYS2*, and the right rDNA repeat loci were replaced by “N” of the same length (*S288c_DSB_chr3_rDNA* reference.”

4. Figure 1I showed how different p value scan alter 4C curves when the d value is set at a fixed number. Unfortunately, the authors did not include scenarios wherein the two parameters can be altered at the same time. Analyses to address this concern should be included. To evaluate the accuracy of the selected parameters under different conditions, the authors should plot the simulated and experimental data side by side within the same graph.

We thank the reviewer for his comments. We expanded upon the polymer model of condensin loop extrusion, presented in a new **Fig. 2E-H**. The full d and p parameter space was sampled and the adequation of the simulation output compared to the experimental data in **Fig. 2F**. The overlay of best-fit simulation and experimental data is shown in **Fig. 2G**. We further determined pause frequency at the centromere in **Fig. 2H** and at the DSB in **Fig. 7C**.

5. In Figures 1-3 and 7, the controls in the 4C plots are based on an average of six loci positioned equidistant from the centromeres as MAT or rDNA. What was the rationale for selecting these specific six loci, and why were additional sites not included? Including more loci with the same criteria—or even all such loci—could provide more robust analyses. Furthermore, how do the authors explain the observed differences between the RE and control curves in regions beyond 200 kb from the viewpoint (e.g., Figure 2)?

The Rab1 organization of budding yeast chromosomes, defined by the clustering of centromeres at the SPB and the attachment of the telomeres at the nuclear envelope, leads to a brush-like organization of chromosome. This brush organization leads sites equidistant from and across the centromeres to interact more frequently than they would if they belonged to the same chromosomal arm. That fact was already noted in the initial 3C paper for chr. III in *MAT α* cells (Dekker & Kleckner, Science 2002).

Brush-like yeast chromosome organization, visualized upon aggregation of the 16 yeast centromeres normalized on random sites along the main diagonal.

The selection of the 6 control sites accounts for this major distortion of the contact distribution along chromosomes by the centromeres. The 6 sites selected most closely match the situation of their associated viewpoints with respect to the chromosome arm length they fall into.

6. In Figure 4C, the minimal differences observed between the Top2-AID and Top2-AID SMC6-AID Nse4/5-AID datasets do not convincingly support the conclusion that Smc5/6 depletion partially relieves the barrier to loop extrusion. Such small variations could easily arise from experimental variability between replicates. To address this, plotting data from three replicates as separate lines or displaying standard deviations is needed to clarify this concern.

This comment is similar to comment 2 by reviewer 1. We agree that a single Hi-C replicate does not allow to ascertain the reproducibility of this effect. Since these experiments originated from an already published manuscript (Jeppsson et al. Mol. Cell 2024), we decided to remove this analysis from the manuscript together with the statement that Smc5/6 is required to block loop extrusion by condensin in a Top2-deficient context.

Since the Smc5/6 datasets show no effect with respect to condensin loop extrusion, we only present the effect of Top2 inactivation/depletion in an updated result section “*Condensin-dependent loops are shortened in the absence of Top2*” (L202-212) and in **Fig. EV4**.

7. Figure 5A-5C lack explanations of what are being plotted, what the “obs” and “exp” data were derived from, what are the “error bars” presented, and how the authors interpret these plots as having or lacking roadblocks for loop extrusion. Figure 5D appears to show a decrease of contacts between RE viewpoint and regions located 20-40kb left of CEN3. Please discuss this change.

We apologize for the confusion and edited Figure 5 to clarify data presentation, as well as the Method section regarding what the “observed/expected” distribution represented in the aggregated contact maps. In brief, the observed contacts in the window surrounding the features of interest (being the 16 centromeres or the *GAL* genes) are divided by the average of same-size windows at random sites along the diagonal.

8. Figure 6A lacks explanations on how detrended contacts were generated; additional details should be provided in the Methods section. The meaning of the red and black lines in the plots is also unclear, as is the rationale for selecting specific enriched sites to overlay with contact profiles instead of others. Additionally, it would be helpful to indicate how statistical significance was assessed in this analysis. Furthermore, PolII peaks do not align well with detrended contact peaks for the rDNA locus (right panels), and the *GAL2* gene shows poor correlation with detrended contact peaks, with the peak located several kilobase pairs away from the *GAL2* gene. The differences between Glu⁺ and Glu⁻ in the bottom panels of Figure 6B-D are subtle; plotting multiple replicates as separate lines or displaying standard deviations can be somewhat helpful.

The Fig. EV6A (former S6A) represents the detrending computation.

We modified the representation of Figure 5 to show the mean and SEM of detrended data smoothed with a kernel of 39 kb (when visualizing large chromosomal regions) or 9 kb when zooming on the *GAL2* gene. This is detailed in the Methods section (L653-671) “*Extraction of intra-chromosomal 4C-like contact profiles*”.

The data are highly consistent across these multiple conditions: an activated *GAL2* stalls condensin loop extrusion while inactive *GAL2* (lactate, lactate+Glu, lactate+Gal+Glu) does not. Indeed, the point of stalling seems to be shifted by a few kilobases relative to the end of the gene. This is not unique to *GAL2* (eg see *AHP1* in **Fig. 5A**). It raises the possibility that local, transcription-induced topological effects stall condensin. Supporting this possibility, Top2 promotes loop extrusion by condensin (**Fig. EV4**).

Other comments:

1. Figure 4C bottom panels, it is unclear whether the orange line represents Top2-AID SMC6-AID Nse4/5-AID/ Top2AID while the blue line represents “SMC6-AID Nse4/5-AID/WT”. Please add descriptions in the figure legend.
2. Figure 4B-4D, Chr III/Chr V ratio is not the most informative comparison. A better control can be the data without DSB induction.

Formation of a DSB leads to the activation of the DNA damage checkpoint that arrests cells in metaphase. In this context the properties of chromatin are modified, most notably through the organization of chromatids as arrays of cohesin-dependent loops (Piazza et al. Nat Cell Biol 2021). Consequently, it is more relevant to compare data of cells in the same physiological state (+DSB at *MAT* or at another site; all arrested in metaphase) than of cells in different states (+ vs - DSB; metaphase vs. asynchronous).

3. Figure 7, RE and DSB appeared to show reduced contact levels at 4h compared with 2h. Does this change reflect the completion of DNA repair? Please explain.

These cells are defective for Rad51 and consequently cannot repair the DSB (L285-287: “Repair-deficient *rad51Δ* mutants were used to (i) ensure the maintenance of a homogeneous population of *MATα* or *MATα* cells (i.e. no switching), and (ii) prevent the Rad51-dependent preferential recruitment of the DSB to the RE in trans (Renkawitz et al, 2013; Dumont et al, 2024).”). This failure to repair can be visualized as a resection tract expanding on each side of the DSB at 2 and 4 hours (new **Fig. EV7A**).

We do not have a definitive explanation for this reduction. It may result from the extensive resection of the break that increases the heterogeneity of the position of the resection front, where condensin likely arrests.

4. “lines 172-173”, G1 $p=120\text{kb}$ is lower than asynchronous cells and metaphase-arrested cells, which does not support the conclusion that “unreplicated DNA and pre-replication complexes are not per se barriers to loop extrusion by condensin.”

This is a good point. However, this decrease is modest (120 kb in G1 vs 150-170 kb in asynchronous and metaphase cells), and no such decrease was observed on chr. XII (170 kb in G1 vs 150-170 kb in asynchronous and metaphase cells). **Fig. 2F** shows that best-fit density/processivity space is not sufficiently sharp to make strong statements about changes in best-fit estimates of processivity of ~30 kb.

However, we agree with the general comment of this reviewer as indeed the depletion of Cdc45 only partly restores loop extrusion on chr. III and XV in S-phase, suggesting that another element than forks inhibits loop extrusion on chromatin in S-phase. We modified the paragraph “Replication forks stall loop extrusion by condensin” as follows (L197-201):

“These observations indicate that replication forks are strong condensin translocation roadblocks, independently of replisome progression. The incomplete restoration of condensin loop extrusion upon Cdc45 depletion further suggests that DDK-primed replisome components can also act as roadblock for condensin, independently of the establishment of replication forks.”

5. “line 251”, it should be Fig S6C not S7C.

Thanks for noticing.

Dr. Aurèle Piazza
Ecole Normale Supérieure de Lyon
Laboratory of Biology and Modelling of the Cell
UMR5239
Lyon 69007
France

12th Feb 2026

Re: EMBOJ-2025-122112

Condensin loop extrusion properties, roadblocks, and role in homology search in *S. cerevisiae*

Dear Aurèle,

Thank you again for submitting a revised version of your transferred manuscript to The EMBO Journal. I sent it back to the original referees 1 and 2, and I am happy to say that they were both satisfied with the revisions and considered the paper substantially improved (as you will see from their comments below). We would therefore like to move forward with publication of the study, as soon as a few remaining editorial issues will have been incorporated:

- Please upload all main Figures and all Expanded View figures as individual files with sufficient resolution/quality for production.
- Please carefully go through the reference list, which contains entries lacking full citation information such as volume or page/eLocator numbers; there also appears to be one reference -Matthey-Doret et al 2020a (possibly a web-tool?)- lacking any sort of citation/DOI/HTML access information.
- For preprint citations: Please change in-text reference format according to "preprint: Miller et al, 2025"; and in the reference list, name of the platform plus DOI + preprint label - e.g. "bioRxiv doi: 1234/002.dfv123 [PREPRINT]"
- In the Data Availability section, please include a specific URL also for the SRA database in which some of the deposited data can be accessed. Furthermore, please note that this section should only mention newly generated datasets or software, while previously established software tools should be listed in the Methods section.
- As we are switching from a free-text author contribution statement towards a more formal statement based on Contributor Role Taxonomy (CRediT) terms, please remove the present Author Contribution section and instead specify each author's contribution(s) directly in the Author Information page of our submission system during upload of the final manuscript. See <https://casrai.org/credit/> for more information.
- In the Disclosure and competing interests statement, please spell out "The authors declare no competing interests with regard to this work".
- Please double-check all figure references in the text, as Fig EV3 currently does not appear to be cited at all, and there is also an incorrect reference to a Figure "S2D-F"
- Please provide suggestions for a short 'blurb' text prefacing and summing up the study in two sentences (max. 250 characters), followed by 3-5 one-sentence 'bullet points' with brief factual statements of key results of the paper; together with the already provided Synopsis image, they will form the basis of an editor-written 'Synopsis' accompanying the online version of the article.
- During routine pre-acceptance checks, our data editors have raised the following queries regarding figures, data, and legends; I would appreciate if you briefly answered to them in the cover letter of your final submission, and made the requested text modifications with changes/additions highlighted via the "Track changes" option, to facilitate our final checking"
 - 1) Please note that information related to n is missing in the legends of figures EV5 D, EV7 F, EV8 C
 - 2) Please note that the error bars are not defined in the legend of figure EV8 C.
- Finally, you shall also receive a separate message from our Source Data curation team, with instructions on how to prepare and upload relevant image and numerical raw data. I appreciate that you already provided some Source Data upon submission, but note that it remains limited to Fig 7D-E at this point.

I am returning the manuscript to you for a final round of minor revision, to allow you to make these remaining modifications and upload the revised files. Once we will have received them, we should be ready to swiftly proceed with formal acceptance and production of the manuscript.

Thank you for the opportunity to consider your work for publication. I look forward to your revision.

With kind regards,

Hartmut

*** PLEASE NOTE: All revised manuscript are subject to initial checks for completeness and adherence to our formatting guidelines. Revisions may be returned to the authors and delayed in their editorial re-evaluation if they fail to comply to the following requirements. As a first step please read our guidelines for revised submissions:
<https://link.springer.com/journal/44318/submission-guidelines#cms-Revised-submissions>

1) Every manuscript requires a Data Availability section (even if only stating that no deposited datasets are included). Primary datasets or computer code produced in the current study have to be deposited in appropriate public repositories prior to resubmission, and reviewer access details provided in case that public access is not yet allowed.

4) Each main and each Expanded View (EV) figure should be uploaded as individual production-quality files (preferably in .eps, .tif, .jpg formats). For suggestions on figure preparation/layout, please refer to our Figure Preparation Guidelines:
<https://media.springernature.com/original/springer-cms/rest/v1/content/27825798/data/v1>

6) Please complete our Author Checklist, and make sure that information entered into the checklist is also reflected in the manuscript; the checklist will be available to readers as part of the Review Process File.

8) Please note that supplementary information at EMBO Press has been superseded by the 'Expanded View' for inclusion of additional figures, tables, movies or datasets; with up to five EV Figures being typeset and directly accessible in the HTML version of the article.

9) To facilitate reproducibility and cross-laboratory adoption of methodologies, please structure the Materials & Methods section as outlined in our guide to authors, including a completed Reagents and Tools Table.

10) Digital image enhancement is acceptable practice, as long as it accurately represents the original data and conforms to community standards. If a figure has been subjected to significant electronic manipulation, this must be clearly noted in the figure legend and/or the 'Materials and Methods' section. The editors reserve the right to request original versions of figures and the original images that were used to assemble the figure. Finally, we generally encourage uploading of numerical as well as gel/blot image source data.

In the interest of ensuring the conceptual advance provided by the work, we recommend submitting a revision within 3 months (13th May 2026). Please discuss the revision progress ahead of this time with the editor if you require more time to complete the

revisions. Use the link below to submit your revision:

Link Not Available

Referee #1:

I read the revised version of the paper, and the authors reply to my original reviews. The authors did a good job with both of my major concerns. First was to remove some of the weak data about smc5/6, and second was to include some functional output data, which they did. I recommend accepting it for publication.

Referee #2:

I appreciate the authors' thorough and thoughtful response to my comments, as well as the substantial additional experiments and analyses they have performed in the revised manuscript.

I find the new experiments addressing the directionality of condensin extrusion at the recombination enhancer very convincing. Most importantly, the authors have now directly addressed the key functional concern I raised regarding the physiological relevance of loop extrusion for donor choice during mating-type switching.

I am satisfied that my concerns have been fully addressed and I now fully support publication of this manuscript.